# Amylin-Calcitonin receptor signaling in the medial preoptic area mediates affiliative social behaviors in female mice

Kansai Fukumitsu [1✉], Misato Kaneko[1,2], Teppo Maruyama[1,2], Chihiro Yoshihara[1], Arthur J. Huang[3], Thomas J. McHugh [3], Shigeyoshi Itohara[4], Minoru Tanaka[2] & Kumi O. Kuroda [1✉]

Social animals actively engage in contact with conspecifics and experience stress upon isolation. However, the neural mechanisms coordinating the sensing and seeking of social contacts are unclear. Here we report that amylin-calcitonin receptor (Calcr) signaling in the medial preoptic area (MPOA) mediates affiliative social contacts among adult female mice. Isolation of females from free social interactions first induces active contact-seeking, then depressive-like behavior, concurrent with a loss of *Amylin* mRNA expression in the MPOA. Reunion with peers induces physical contacts, activates both amylin- and Calcr-expressing neurons, and leads to a recovery of *Amylin* mRNA expression. Chemogenetic activation of amylin neurons increases and molecular knockdown of either amylin or Calcr attenuates contact-seeking behavior, respectively. Our data provide evidence in support of a previously postulated origin of social affiliation in mammals.

[1] Laboratory for Affiliative Social Behavior, RIKEN Center for Brain Science, Saitama 351-0198, Japan. [2] Department of Animal Science, Faculty of Applied Life Science, Nippon Veterinary and Life Science University, Musashino, Tokyo 180-8602, Japan. [3] Laboratory for Circuit and Behavioral Physiology, RIKEN Center for Brain Science, Saitama 351-0198, Japan. [4] Laboratory for Behavioral Genetics, RIKEN Center for Brain Science, Saitama 351-0198, Japan. ✉email: kansai.fukumitsu@riken.jp; kumi.kuroda@a.riken.jp

Loneliness or perceived social isolation has long been known as a major risk factor for diseases and mental disorders, including depression, cognitive decline, obesity, cancer, heart disease, and even premature mortality in humans[1,2]. Adults of sociable, or non-solitary species[3] display a preference for affiliative contact distinct from that serving reproductive or agonistic/competitive purposes (see Supplementary discussion for term definition). Nonhuman mammals with sociable traits, including laboratory rats, mice, and the majority of diurnal primates, live in groups and actively engage in social contact at least under certain conditions[3–5]. When separated from peer conspecifics these sociable animals may exhibit a two-stage response, with initially enhanced and later depressed physical activity and stress responses, such as activation of the hypothalamic-pituitary-adrenal (HPA) axis[6–9]. Alternatively, there are beneficial effects of a supportive social environment (termed "social buffering") on coping with adverse conditions, such as environmental stress and diseases[10,11]. However, while numerous studies have investigated the long-term effects of prolonged social isolation (for example, Zelikowsky et al.[12]), less is known about the early responses to social isolation.

Moreover, the great majority of research on social contacts and isolation has focused only on male animals[12–18]. Compared with females, however, male-male encounters tend to be more agonistic in both feral and laboratory conditions[19]. Therefore, it is not surprising that male mice may exhibit fewer isolation-induced depressive behaviors than females and rather, can show stress responses to group housing[20–22]. Further, the agonistic/competitive nature of male social interaction is not well captured by measures of indirect social interaction, such as the three-chambered social preference test[23], in which automated scoring of proximity to a social versus a nonsocial chamber can lead to misinterpretation of the subject animals' sociability.

In contrast, in many mammalian species, females exhibit higher sociability and emotional responses to social isolation than males[3,21,24–27]. This may be related to maternal care: group living often provides significant benefits for females' offspring care. The existence of social bonds between adult females have been shown to enhance infant growth and survival in many mammals, including humans[28,29], baboons[30], and mice[19,31]. Taylor and colleagues have proposed that a "tend-and-befriend" strategy with conspecifics may be better suited for coping with stress in females with offspring than a "fight-or-flight" response, as fighting or escaping is risky for the young[32]. Taken together, as previously suggested, it is not only beneficial but also indispensable, to use female subjects to elucidate the neural mechanism of sociability and social affiliation[25,33].

Here we investigated the neural circuits underlying the sensing and seeking of social contacts using adult female laboratory mice. We found that the medial preoptic area (MPOA), especially the central part (cMPOA) which lies in the posterior MPOA, is essential for parental and alloparental care in rodents[34–37]. Specifically, cMPOA neurons expressing the Calcitonin-receptor (Calcr) play a key role in these behaviors[38]. Calcr is a G-protein coupled receptor, which forms a complex with Receptor Activity Modifying Proteins (Ramps) to bind amylin in the brain[39]. Amylin, a 37-amino acid bioactive brain-gut peptide, also called as IAPP (islet amyloid polypeptide), is co-secreted with insulin from pancreatic β cells and inhibits food intake[40,41]. Amylin is also produced in the hindbrain, lateral hypothalamus, arcuate nucleus, and MPOA[42] and functions in nociception and maternal adaptation[43–45]. While studying Calcr+ and amylin+ neurons with regard to maternal care, we observed that amylin expression in MPOA subregions is heavily dependent on social housing conditions in female mice. Therefore, in this study we dissected the molecular and circuit underpinnings of these socially induced

changes in female mice by means of genetic knockdowns, amylin infusion, and chemogenetic DREADDs experiments[46], and found that amylin-Calcr signaling in the cMPOA is required for contact seeking behavior in an affiliative context.

## Results

### Social isolation abolishes *Amylin* mRNA expression in the MPOA.
Among the whole MPOA, amylin is robustly expressed in the cMPOA and the anterior commissural nucleus (ACN) and is only moderately expressed in the medial part of the medial preoptic nucleus (MPNm) of C57BL/6 adult female mice under group housing conditions (Fig. 1a, b). However, *Amylin* mRNA expression in these MPOA subregions was reduced by half following 2 days of single housing (isolation in the home cage by removing all cage mates), and became undetectable by 6 days of isolation (Fig. 1c-e). Subsequent cohousing led to a gradual recovery of *Amylin* expression to original levels, but if isolation continued, the expression level remained at zero. Cohousing with four unfamiliar females or with four castrated males was equally effective in maintaining *Amylin* mRNA expression, but cohousing with one unfamiliar female was less effective (Fig. 1f-h). Similar regulation of MPOA *Amylin* expression by housing conditions was observed in BALB/c females (Supplementary Fig. S2e-h). These results suggested a dependence of *Amylin* mRNA expression on the amount of social interaction.

### *Amylin* mRNA expression depends on social contact but not on stress conditions.
Social isolation is stressful for female laboratory mice and rats[21,47–49]. Therefore, *Amylin* mRNA expression could be sensitive to general stress rather than to social conditions per se. Thus, we subjected female mice to restraint stress with or without conspecifics in a two-by-two design (Fig. 1i). This experiment demonstrated that *Amylin* mRNA expression was regulated by social condition, but not by the presence or absence of restraint stress (Fig. 1j).

The monotony or lack of stimulation during isolation may also impact *Amylin* mRNA expression. However, neither environmental enrichment by providing attractive non-social stimuli (e.g., a running wheel, novel objects as a toy), nor different palatable foods every day were sufficient to induce *Amylin* mRNA expression in single-housed females (Supplementary Fig. S1a-c).

Moreover, using the immediate-early-gene product c-Fos as a readout of transcriptional activation of neurons, we found that amylin-ir (immunoreactive) neurons were activated by 2 h of co-housing after isolation (Fig. 1k-m). Furthermore, amylin-ir neurons showed continuous activation during group housing, and were de-activated within 2 h of isolation (Fig. 1l, n). These data collectively indicated that amylin-ir MPOA neurons are activated by the presence of social stimuli and that such activation is associated with an increase amylin expression level within the MPOA.

### Free social interaction is necessary to induce *Amylin* mRNA expression.
The sensitivity of amylin expression to the social housing conditions suggests that sensory cues from conspecifics might be required to maintain amylin expression. Because mice are macrosmatic and depend largely on pheromones and volatile chemosignals for social behaviors[50–52], we first tested the role of olfaction. However, the daily addition of soiled bedding of cage mates, which contained both volatile and nonvolatile chemical signals from peers, was not sufficient to maintain *Amylin* expression (Fig. 1o). Presenting a single-housed female mouse with a cold or warmed mock mouse, an anesthetized cage mate, or an intact female mouse for 3 h per day x 4 days also failed to maintain *Amylin* expression in the MPOA (Fig. S1e).

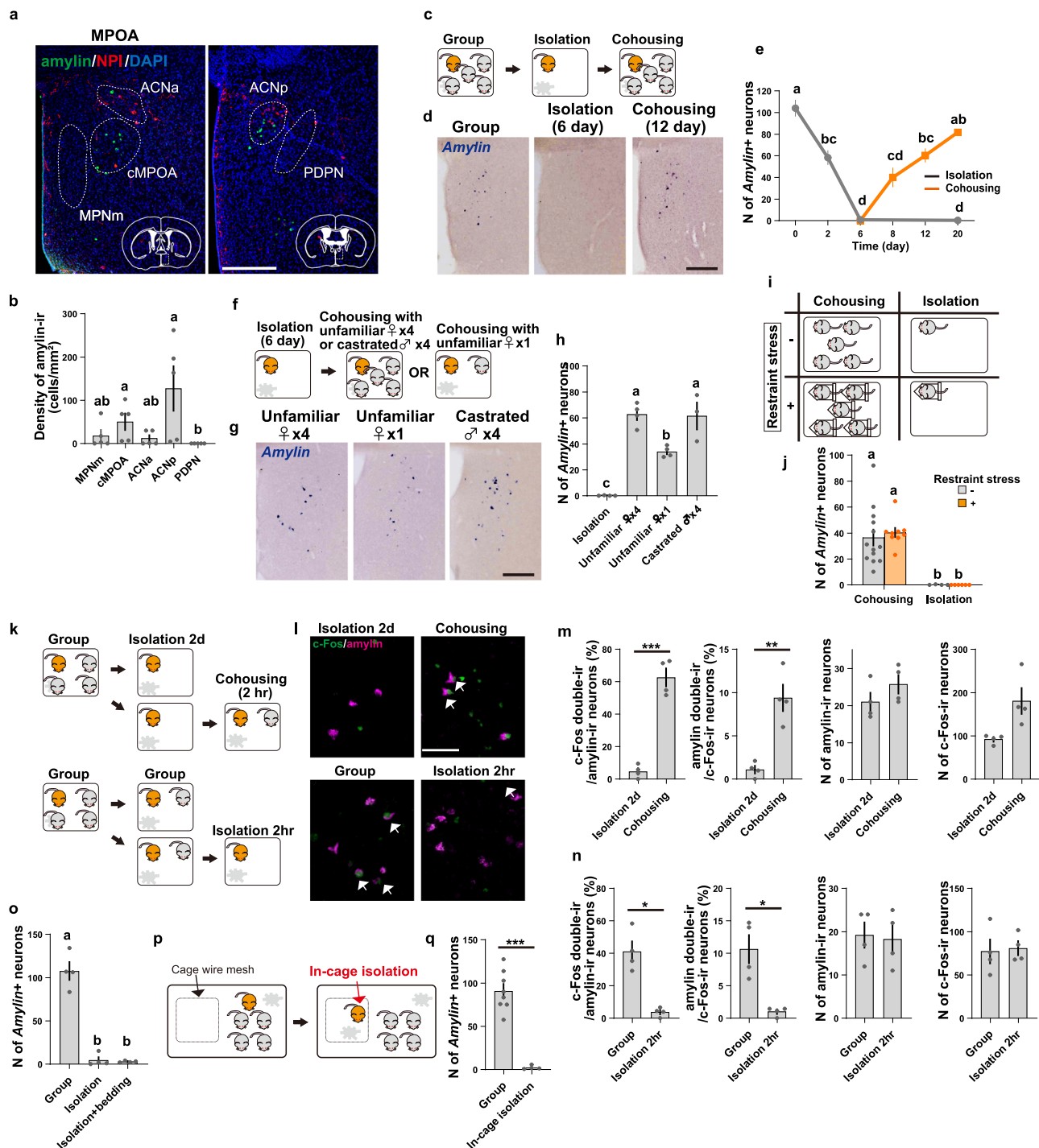

To understand the sensory information involved in maintaining *Amylin* expression we next segregated a single female in a wire-mesh compartment within a large group cage. Identical to the effect of single-housing, in-cage isolation resulted in a loss of *Amylin* expression after 6 days (Fig. 1q, Supplementary Fig. S1f). Thus, the olfactory, visual, auditory, and limited tactile inputs that are available during this in-cage isolation are not sufficient to maintain amylin expression in the MPOA, suggesting that free social interaction is required.

**The cage system that enables simple manipulations of housing condition to social isolation and reunion**. To further investigate neural circuit supporting affiliative social behavior among adult

female mice we designed experiments employing MPOA amylin expression as a molecular readout of isolation. We first performed several preparatory behavioral experiments and shifted the animal model from C57BL/6 to BALB/c adult female mice. BALB/c adult females exhibited clearer responses to social isolation and reunion, lacking atypical mounting behavior of introduced females observed in C57BL/6 females which complicates behavioral scoring (Fig. S1g-i). Importantly, the amylin expression pattern in response to isolation described in the C57BL/6 mice was identical in the BALB/c female mice (Fig. S2e-h).

In addition, we implemented a simpler version of in-cage isolation system (Fig. 2a). One of two types of partition walls were set in the middle of a large test cage, one a sham partition with two large windows that mice can go through (Fig. 2a, bottom left)

**Fig. 1 *Amylin* mRNA expression in the MPOA depends on direct social contacts. a** Distribution of amylin-immunoreactivity (ir, green) neurons, NPI-ir (red) and DAPI staining (blue) in two coronal brain sections (left: anterior, right: posterior) from a group-housed female mouse. The conservative contours of the MPOA subregions are overlayed. Scale bar, 300 μm. **b** Density of amylin-containing neurons in each subregion ($n = 5$ mice per group). (**c–e**) Social isolation abolishes and cohousing increases *Amylin* mRNA expression in the MPOA. **d**, **g** *Amylin* mRNA detected by (ISH, blue) in the MPOA. Scale bar, 250 μm. **e** Quantification of the number (N) of MPOA neurons expressing *Amylin* mRNA, 0-day, 6-day: $n = 7$ mice, 2-day: $n = 10$, other groups: $n = 6$. (**f–h**) Cohousing with four unfamiliar females or with four unfamiliar castrated males effectively maintained *Amylin* mRNA expression. (**h**) Quantification of the number of neurons expressing *Amylin* mRNA in the MPOA (castrated male group: $n = 3$ mice, other groups: $n = 4$). (**i**, **j**) Repeated restraint stress did not affect *Amylin* mRNA expression in the MPOA. (**j**) Quantification of the number of neurons expressing *Amylin* mRNA in the MPOA. (isolation + non-stressed group: $n = 4$, isolation+stressed group: $n = 6$, group+non-stressed: $n = 13$, group+stressed group: $n = 9$). (**k–n**) Activation of amylin-ir neurons after isolation or cohousing. (**k**) Top: group-housed female mice were isolated for 2 days and then cohoused for 2 h or kept isolated. Bottom: female mice housed in group were isolated for 2 h or kept in group. (**l**) Distribution of amylin-ir (magenta) and c-Fos-ir (green) neurons in the MPOA. Arrows: double-labeled cells. Scale bar, 50 μm. (**m, n**) Number of amylin-ir and c-Fos-ir neurons, percentage of amylin-ir neurons among the c-Fos-ir neurons, percentage of c-Fos-ir neurons among the amylin-ir neurons in the MPOA ($n = 4$ mice per group). (**o**) Chemosignals from cage mates had no impact on *Amylin* mRNA expression. Soiled bedding from cage mates was introduced daily to the cages of isolated mice ($n = 4$ mice per group). (**p, q**) In-cage isolation depleted *Amylin* mRNA expression (group-housed group: $n = 8$ mice, in-cage isolation group: $n = 4$). See Supplementary Fig. S1f for details. Asterisks indicate significant differences between two groups (Welch's unpaired t-test; *$p < 0.05$, **$p < 0.01$, ***$p < 0.001$). The letters indicate significant differences (**b**: Kruskal-Wallis test with Benjamini-Krieger FDR, **e**: Welch's ANOVA with Dunnett's T3 multiple comparison test, two-sided, **h**, **o**: one-way ANOVA with Tukey's multiple comparison test, **j**: two-way ANOVA with Sidak's multiple comparison test, $p < 0.05$). Graphs show mean ± SEM. See Supplementary Data for exact *p* values and details of statistical analyses.

but otherwise similar to the real partition; the other a real partition with two large barriered windows that allowed for visual, auditory, olfactory, and even some direct contact, but prevented the passage of the mice (Fig. 2a, bottom right). The isolation across this real partition was regarded as equivalent to the preliminary in-cage isolation condition shown in Fig. 1p and Supplementary Fig. S1f. Moreover, with this new apparatus, changing social conditions did not require stressful mouse handling, which can induce alarm calls and/or urination, leading to stress responses even in the non-handled animals[53]. Hereafter, to specify different kinds of isolation, we termed single housing in one cage as *Complete isolation* (Fig. 2b), and use *Somatic isolation* to describe a subject mouse housed in the same cage with other mice but separated by a coarse mesh or windowed partition as in Fig. 2a and Supplementary Fig. S1f. *(Social) isolation* includes both complete and somatic isolations.

**Somatic isolation induces contact-seeking behavior in female mice.** To study the behavioral and physiological responses to social isolation and reunion, four BALB/c adult female mice, which were group-housed from weaning, were placed in a cage with a sham partition, and habituated for 7 days. Then the sham partition was replaced with a real partition to create four different social conditions (*4-together*: all four mice were enclosed into a no-nest chamber, *Complete isolation*: one mouse was caged into a no-nest chamber and the other mice were removed from the cage, *3-together*: three mice staying in the chamber with their nest, *Somatic isolation*: one of the mice was separated into a chamber while three cage-mates with their nest were on the other side of the real partition (Fig. 2b). The mice in each group were monitored by video, as well as an implantable locomotion monitor (Nano Tag™). Mouse behaviors after the partition replacement were coded based on previously-reported mouse ethogram[54,55] as follows: 1) General movements: still (no movement), movement (locomotion or any physical movements not otherwise classified); 2) exploration of the environment: sniffing (targeted at the partition or through the partition), rearing, digging; 3) contact-seeking: biting at the partition, 4) non-social behaviors: self-grooming, eating, panic.

Immediately after introduction of the real partition the general activity of all the groups significantly increased (Fig. S2a). The increased activity was mainly targeted toward the new partition, including walking around/sniffing the partition or sniffing while inserting their nose into the slit of the windows (Fig. 2c),

indicating that all the mice investigated the altered environment. After 45 min, these exploratory behaviors decreased, with no statistically significant differences in total locomotor activity or core body temperature among the groups (Fig. S2a-d). However, the detailed behavioral analysis revealed that during the initial 30 min after introduction of a real partition, mice under complete and somatic isolation demonstrated more exploratory rearing and digging behaviors around the partition than mice in 4- and 3-together conditions (Fig. 2c, illustrated as the left-most, shaded part within each bar). In addition, somatically-isolated mice bit the partition significantly more than mice in the other groups (Fig. 2d-f). In contrast, complete isolation induced more self-grooming (Fig. 2c), which often occurs as a displacement behavior under stressful stimuli in many mammals, including mice[56]. 3- and 4-together groups did not differ in any variables throughout this experiment, suggesting that these isolation-induced behaviors were not because of separation from the nest. Taken together, partition-biting was increased when the subject mice were isolated and inaccessible peers were present beyond the partition, suggesting that the partition-biting may represent social motivation or contact-seeking behavior.

**Inactivity in a hunched posture as the second-stage reaction to social isolation.** Following the initial 90 min, mice in complete and somatic isolation exhibited a specific hunched posture (Fig. 2g) significantly more than the mice in nonisolated, 4- or 3-together groups (Fig. 2h-i). This hunched posture is distinct from a crouch, a behavior similar across groups in which a mouse stands in all four limbs with relaxed neck and spine, so the snout points forward (Fig. 2j-l); in a hunched posture, a mouse stands still on the hind limbs with deeply flexed neck, so that the snout points toward their ventral trunk, and typically eyes closed (Fig. 2g), similar to that described in isolated guinea pigs[57], with the exception of piloerection which is difficult to detect in mice.

With respect to a stress response, 120 min after the introduction of the real partition mice in complete and somatic isolation exhibited a significant increase in c-Fos expression in the paraventricular nucleus of the hypothalamus (PVH), the central regulator of endocrine stress responses, compared to mice in the 3- or 4-together conditions (Fig. 2m-n). These findings suggest that both complete and somatic isolation are stressful for female mice, but lead to distinct coping behaviors.

Somatic isolation induced a two-stage response, consisting of the initial active phase of contact-seeking and partition biting for

**Social isolation**

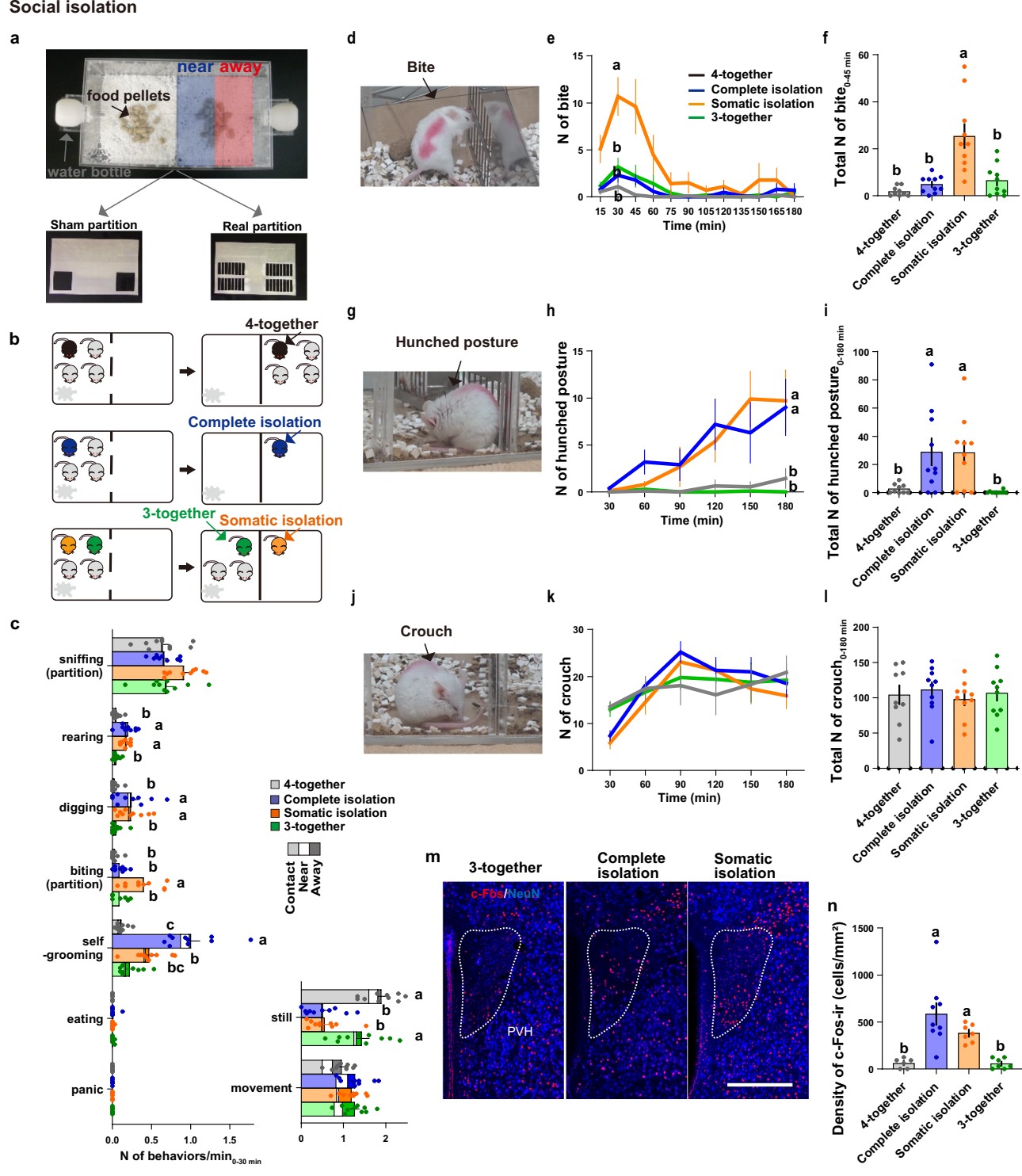

about 45 min, followed by the following inactive phase of hunched posture. This two-staged reaction is similar to those originally documented in maternally separated young of humans, nonhuman primates and guinea pigs, and also more generally, in stressed animals[9,58–61]. However, peer-separated adult female mice did not show panic reaction (Fig. 2c) nor elevation of body-core temperature (Fig. S2c-d), possibly because the peer separation of adult mice is not as threatening as maternal separation of infants. To identify the neural mechanisms of sensing and seeking for social contacts we next focused on the initial active response to *somatic isolation*, which induces more

contact-seeking behavior and retains the visual, auditory, and olfactory signals from the cage-mates, eliminating complications in interpreting changes in behavior or neuronal activities.

**Characterization of two kinds of social contacts: reunion and defensive huddle.** Animals under somatic isolation for 2 days were next subjected to a "reunion" procedure; the real cage partition was replaced with a sham partition (Fig. 3a), and the behaviors of previously-isolated mice were coded for the above-described categories 1)~4), as well as 5) direct social behaviors:

**Fig. 2 Social isolation induces a two-stage response in female mice. a** The experimental cage used for social isolation-reunion test. A sham (left) or real (right) partition was placed in the middle of the cage. The blue and red areas indicate the mouse positioning in the cage analyzed in (**c**). **b** Four different social conditions were created by replacing the partition as explained in the text. (**c**) Quantification of behaviors, coded at 15-s intervals for 30 min under different social conditions ($n = 10$ mice per group). The distribution of the positions at which mice performed the behavior is represented by the bar shading (light, none, dark). The letters indicate significant differences (Welch's ANOVA with Dunnett's T3 multiple comparison test, two-sided for partition-biting, Welch's ANOVA with Benjamini and Krieger FDR test, two-sided for digging, and one-way ANOVA with Tukey's multiple comparison test for other behaviors, $p < 0.05$). (**d–l**) Analysis of the partition-biting behavior (**d–f**), hunched posture (**g–i**), and crouch (**j–l**) after adding the real partition. Pictures of typical behaviors (**d, g, j**), time course, and total number (N) of behaviors recorded at 15-sec intervals during 45 min (**e–f**) for biting and during 180 min for the hunched posture (**h–i**) and crouch (**k–l**) (4-together group: $n = 9$, other groups: $n = 10$ mice). The letters indicate significant differences (Welch's ANOVA with Dunnett's T3 multiple comparison test, two-sided in (**e**) and (**f**), Kruskal-Wallis test with Benjamini-Krieger FDR in (**h**), and Kruskal-Wallis test with Benjamini-Hochberg FDR in (**i**), $p < 0.05$). (**m, n**) PVH neuron activity assessed by c-Fos (red) and NeuN (blue) immunostaining. Scale bar, 250 µm. Somatic isolation and 3-together groups: $n = 7$ mice, 4-together group: $n = 6$, and complete isolation group: $n = 9$. The letters indicate significant differences (one-way ANOVA with Tukey's multiple comparison test, $p < 0.05$). Graphs show mean ± SEM. See Supplementary Data for exact $p$ values and details of statistical analyses.

crawling under/pushing, sniffing (peer), allogrooming, mounting, biting (peer), chasing. Reunion induced general movement, digging, and peer-sniffing during an initial phase lasting 10–15 min (Fig. 3b). In BALB/c female mice no agonistic or sexual behaviors were observed toward peer females. Following this phase, the female mice gradually became inactive and formed a cluster termed a sleep huddle roughly 30 min after reunion (Fig. 3c-d). It should be noted that while sleep huddles have an obvious role in thermoregulation at low temperatures, they also involve social tolerance and are facilitated by familiarity[62,63].

For comparison, here we also characterized a distinct situation that involves significant physical contact among cage mates, called a *defensive huddle*[64]. A defensive huddle is defined as aggregation of individual animals as a strategy of self-defense under a potential threat. This primitive gregarious behavior is designated as a "selfish herd"[65] and thus does not have an affiliative nature toward conspecifics. To induce defensive huddle behavior in BALB/c females we utilized their sensitivity to bright light[66]. Two adult females were habituated to the testing room and then were transferred to a novel cage either in dark or under bright light during the dark phase (Fig. 3e). The females under bright light exhibited significantly more social contacts in both total frequency and duration than those in dark, along with more defecation as a behavioral sign of stress (Fig. 3f-g). Moreover, we observed a significant increase in the number of PVH c-Fos+ neurons following 30 minutes-bright illumination (Fig. 3h-i), but not after 2 h of reunion (Fig. 3h-i). Therefore, while reunion and light-induced defensive huddle both involve increased physical contacts with cage mates, these two conditions have distinct features, and reunion does not include a stress response.

**Brain activity mapping during somatic isolation, reunion, and defensive huddle.** We next investigated the brain regions activated during the above-described social conditions. For the social and emotional behaviors, previous literature suggests the importance of the extended amygdala, including the bed nuclei of stria terminalis (BST)[67,68] (see also[69–71]). The BST is of additional interest as it is adjacent to the amylin-expressing MPOA. Thus, here we investigated neuronal activation patterns in the subregions of MPOA, BST, and amygdala during somatic isolation, reunion, and bright-light induced defensive huddle. Specifically, we identified five MPOA subregions (Fig. 4a, left) based on our previous anatomical study[36], the MPNm and the posterodorsal preoptic nucleus (PDPN) are characterized by the dense estrogen receptor alpha (ERα)-ir neurons[72]. The ACN is characterized by the cluster of magnocellular oxytocin neurons[73], which was detected by neurophysin I (NPI), the cleavage product of pre-prooxyphysin that acts as a carrier protein of oxytocin. The cMPOA is between MPNm and ACN[36]. For the BST and

amygdala, five and ten subregions were identified, respectively, using immunostaining of neuronal nuclei (NeuN) (Fig. 4a, middle and right)[37,74].

Using c-Fos as a readout we observed that two hours after somatic isolation neurons were activated in the central nucleus of amygdala, medial and lateral part (CeM and CeL), the anterior amygdaloid area (AA), and the rhomboid and anterolateral part of BST (BSTrh and BSTal) (Fig. 4c, S2i). BSTrh and BSTal are heavily connected with CeM[75–77] and are known for their roles in fear and anxiety. In contrast, we did not observe any significant changes of c-Fos expression in the preoptic subregions (Fig. 4c).

Reunion with three cage-mates induced robust activation of cells in the preoptic area, specifically in the MPNm, cMPOA, ACNa, as well as in the principal nucleus of BST (BSTpr) and a part of amygdala subregions such as the anterodorsal and posterodorsal parts of the medial amygdala (MeAD and MePD) (Fig. 4e, S2i).

To identify the regions associated with light-induced defensive huddle, two mice were transferred into a novel cage in dark or brightly illuminated light. 30 min later, they were returned to the home cage and subjected to immunohistochemical studies 1.5 h later (Fig. 4f, g). The baseline c-Fos levels were determined in the condition where two mice were left undisturbed in their home cage (Fig. 4f). Transfer to a brightly lit novel cage induced significant increases in the number of c-Fos in the BSTpr, BSTal, MeAD, AA, the capsular part of the central amygdala (CeC) and the basolateral amygdala (BLA) in comparison with baseline (Fig. 4g), while the transfer to a dark novel cage did not induce statistically significant changes in any brain regions.

In summary, no overlap was found between the brain regions activated by the reunion and those activated by somatic isolation. The BSTpr and MeAD were commonly activated after reunion and bright-light induced defensive huddle, suggesting these regions might be activated in response to direct physical contact. On the other hand, the preoptic subregions, namely the ACN, cMPOA, MPNm were activated only by reunion, suggesting that these regions were associated specifically with affiliative contacts.

**Calcr expressing neurons in the cMPOA are activated during reunion.** Amylin binds and activates the calcitonin receptor (Calcr) when complexed with one of three receptor activity–modifying proteins (RAMP 1-3)[78]. Our previous study has shown expression of Calcr and RAMP2 in cMPOA and ACN neurons, which are distinct from neurons expressing amylin[38], suggesting that local amylin release may act on Calcr-expressing (Calcr + ) neurons. In the forebrain of BALB/c female mice Calcr+ neurons were found in the ACN, cMPOA, MPNm, BSTpr, and MePD (Fig. 5a, b). This spatial pattern of Calcr expression in the preoptic-strial-amygdala area showed notable

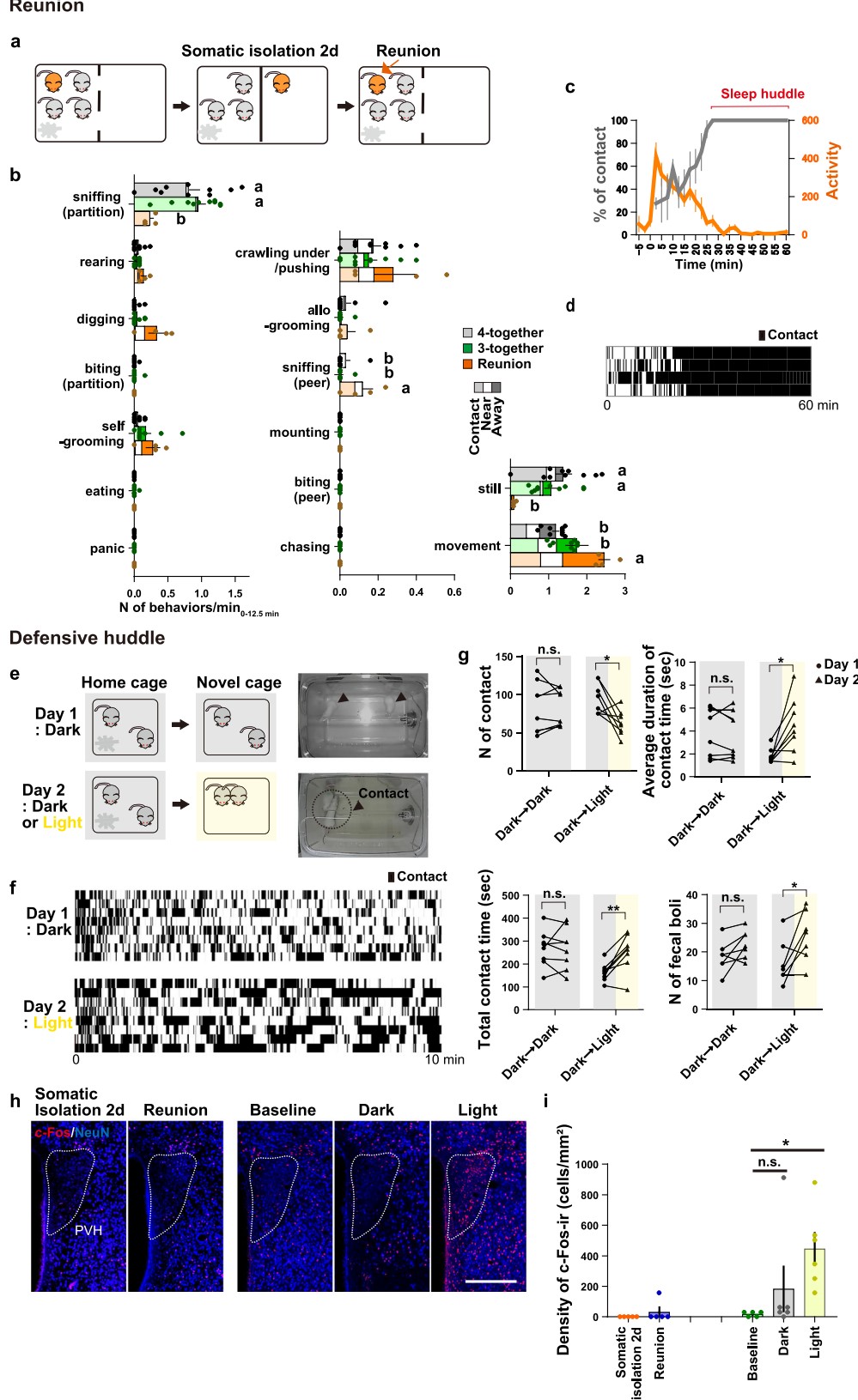

similarities with the pattern of neuronal activation specifically by reunion (Fig. 4e). The localization of amylin+ neurons also clearly coincided with the neuronal activation pattern by reunion in the MPOA (Fig. 1a, b), but the numbers of amylin+ neurons were much more limited, with no amylin+ neuronal soma observed in the BST or the MeA (see also[42]), thus it is

possible that amylin+ neurons may send projections from the MPOA and act on Calcr+ neurons in these other areas.

Following reunion, we found a significant increase in the co-localization of Calcr and c-Fos in the cMPOA and MePD compared to controls (Fig. 5c-d), but not in the ACNa or ACNp, even though both reunion-induced c-Fos expressing and Calcr+

**Fig. 3 Characterization of two types of social contacts: reunion and defensive huddle. a** Social reunion procedure. **b** Quantification of various behaviors, coded at 15-sec intervals for 12.5 min, after social reunion in the 3- or 4-together conditions. The distribution of the positions at which mice performed a given behavior is represented by the bar shading (light, none, dark). Reunion group: $n = 4$ mice, 4-together and 3-together groups: $n = 10$. The letters indicate significant differences (Welch's ANOVA with Benjamini and Krieger FDR test for analyzing the peer-sniffing and one-way ANOVA with Tukey's multiple comparison test for the other analyses, $p < 0.05$). (**c**) Time course analysis of social contact (gray) and locomotor activity (orange) after reunion. Time bins: 30 s (locomotion) and 15 s (contact), $n = 4$ mice. **d** Raster plots of social contact bouts. **e** Left panel: light-induced defensive huddle procedure. Two female mice were transferred from their home cage to the dark/light novel cage during the dark period (Day 1: dark novel cage, Day 2: dark or bright novel cage). Right panel: representative pictures of the control dark condition (top) and the defensive huddle seen in the stressful light condition (bottom). **f** Raster plots of mouse social contact bouts in the dark or light conditions. (**g**) Number (N) of contacts, total contact time, average duration of the contacts, and number of fecal boli determined in each condition after 10 min ($n = 8$ mice per group). Asterisks indicate significant differences between two groups (paired $t$-test; *$p < 0.05$, **$p < 0.01$). Videos recorded continuously for 10 min were used for analysis. (**e**–**g**). (**h**) PVH neuron activity measured by c-Fos (red) and NeuN immunostaining (blue). Scale bar, 250 μm. (**i**) Density of c-Fos-immunoreactive (c-Fos-ir) neurons in the PVH after distinct social stimulations (dark and light condition: $n = 6$ mice, 2 days (2d) somatic isolation, reunion, and baseline conditions: $n = 5$). Astarisks indicate significant differences between baseline and light groups (one-way ANOVA with Dunnett's multiple comparison test, two-sided, *$p < 0.05$). Data represent mean ± SEM. See Supplementary Data for exact $p$ values and details of statistical analyses.

neurons were the densest in this subregion, suggesting a functional distinction of Calcr+ neurons in the cMPOA and ACN. Indeed, while the most ACN Calcr+ neurons expressed *GAD67* mRNA, a marker for GABAergic neurons in this region, the majority of cMPOA Calcr+ neurons expressed *VGlut2* mRNA, an index for glutamatergic neurons (Supplementary Fig. S2j). In contrast with the reunion, Calcr+ neurons were not activated by defensive huddle nor somatic isolation in all brain areas examined (Fig. 5e-f). We confirmed scarce activation even by more extensive physical contact with a peer caused by defensive huddle for predator-odor analog 2-methyl-2-thiazoline (Supplementary Fig. S3). Taken together, these data suggest Calcr+ neurons in the cMPOA and MePD are involved in affiliative, but not defensive, social contacts.

**Knockdown of Calcr expression in the cMPOA blocks behavioral responses to somatic isolation and reunion.** Social reunion robustly activates Calcr+ neurons in the cMPOA. As *Calcr* gene knockout mice die as embryos[79], to address if the calcitonin receptor is required for the behavioral response to somatic isolation and/or reunion we utilized AAV-mediated short hairpin RNA interference (shRNA; AAV5-*hH1-shCalcr-CAG-EGFP*) to knockdown Calcr, and an AAV5-*hH1-Scrambled-CAG-EGFP* as a control (Fig. 6, S4). This method has been previously shown to result in the knockdown of Calcr expression in the ventral tegmental area[80] and in the MPOA[38]. Infection of both viral vectors was confirmed to cover at least two-third of the bilateral cMPOA (Fig. S4). We confirmed that the shRNA targeting *Calcr*, but not the scrambled control RNA, significantly reduce Calcr expression in the cMPOA two weeks after injection (Fig. 6a-c), observing a 74.3 % reduction of Calcr immunostaining (Fig. 6b). Fluorescent immunohistochemical analysis revealed that in the shRNA-injected cMPOA, neurons did not colocalize EGFP with Calcr, while in the cMPOA injected with the scrambled RNA, overlap of EGFP and Calcr expressions were observed, indicating that only the shRNA but not the scrambled RNA inhibited Calcr expression (Fig. 6a).

BALB/c female mice injected with *Calcr* shRNA mice in the cMPOA exhibited attenuated somatic isolation-induced rearing, digging (Fig. 6d) and biting behaviors (Fig. 6e-f). Upon reunion, Calcr-knockdown in the cMPOA also reduced crawling-under/pushing (Fig. 6k) and contact behaviors (Fig. 6g-j) and increased unclassified movements and self-grooming (Fig. 6k) during the initial phase. However, Calcr knockdown in the cMPOA did not inhibit long-term contact behaviors or sleep huddling (Fig. 6l-n).

We also confirmed the role of the cMPOA in the behavioral responses to somatic isolation and reunion using N-methyl-D-aspartic acid (NMDA) excitotoxicity, which eliminate neurons

expressing NMDA receptors but spares passing fibers (Fig. S5-S6). cMPOA lesions reduced partition-biting after somatic isolation and initial contact after reunion to levels similar to that observed in the Calcr knockdown mice, as well as reducing sleep huddle behavior. These data indicate that the cMPOA functions in inducing contact-seeking and the acute phase of affiliative contact behaviors were largely mediated via Calcr in the cMPOA.

**Amylin activates Calcr+ neurons in the cMPOA.** The data above suggest that social activity activates amylin+ neurons and maintains amylin expression, which can subsequently signal to Calcr+ neurons to facilitate further social contact. To address whether and how Calcr+ and amylin+ neurons interact in the cMPOA we first investigated the morphological relationship between these neurons. Anti-Calcr immunohistochemistry visualizes the entire plasma membrane, including neuropils, of Calcr+ neurons (Fig. 7a), however, like other bioactive neuropeptides, subcellular localization of amylin depicted by an anti-amylin antibody was found mainly in the cis Golgi complex, as seen in the co-localization of amylin with the cis-Golgi matrix protein GM130 (Fig. 7b). Thus, to visualize the neuropils of amylin+ neurons, *Amylin-Cre* transgenic mice were generated (Fig. S7) and were injected with AAV2-hSyn-DIO-EGFP in the cMPOA. Calcr+ synaptic terminals were often located in close proximity to the soma of amylin+ cells (Fig. 7c-d), however, we could not detect Amylin+ synaptic terminals near the soma of Calcr+ cells (Fig. 7d). These data suggests that Calcr+ neurons may be presynaptic, with amylin signaling back to Calcr+ neurons. This type of retrograde neuromodulation has been previously seen with neuropeptides, and in the opioid, anandamide, or nitric oxide systems[81,82]. Nonetheless, to test if amylin could bind to Calcr and regulate Calcr+ neurons, amylin was locally infused into the cMPOA via cannula, resulting in a significant c-Fos induction in Calcr+ neurons (Fig. 7e-h). Therefore, amylin is indeed the ligand of Calcr in the cMPOA and activates Calcr+ neurons.

**Amylin-Calcr signaling induces contact-seeking behaviors.** To further elucidate the function of amylin in social contact-related behaviors, we next examined the behavioral effect of chemogenetic activation of amylin+ neurons. *Amylin-Cre* female mice were unilaterally injected with a virus expressing the Cre-dependent *DREADD* neuronal activator, hM3Dq (AAV5-hSyn-DIO-hM3(Gq)-mCherry)[46], into the cMPOA. After two weeks of group housing, the subject mice received a subcutaneous injection of a hM3Dq agonist clozapine-N-oxide (CNO) and were then subjected to somatic isolation for two hours (Fig. 8a).

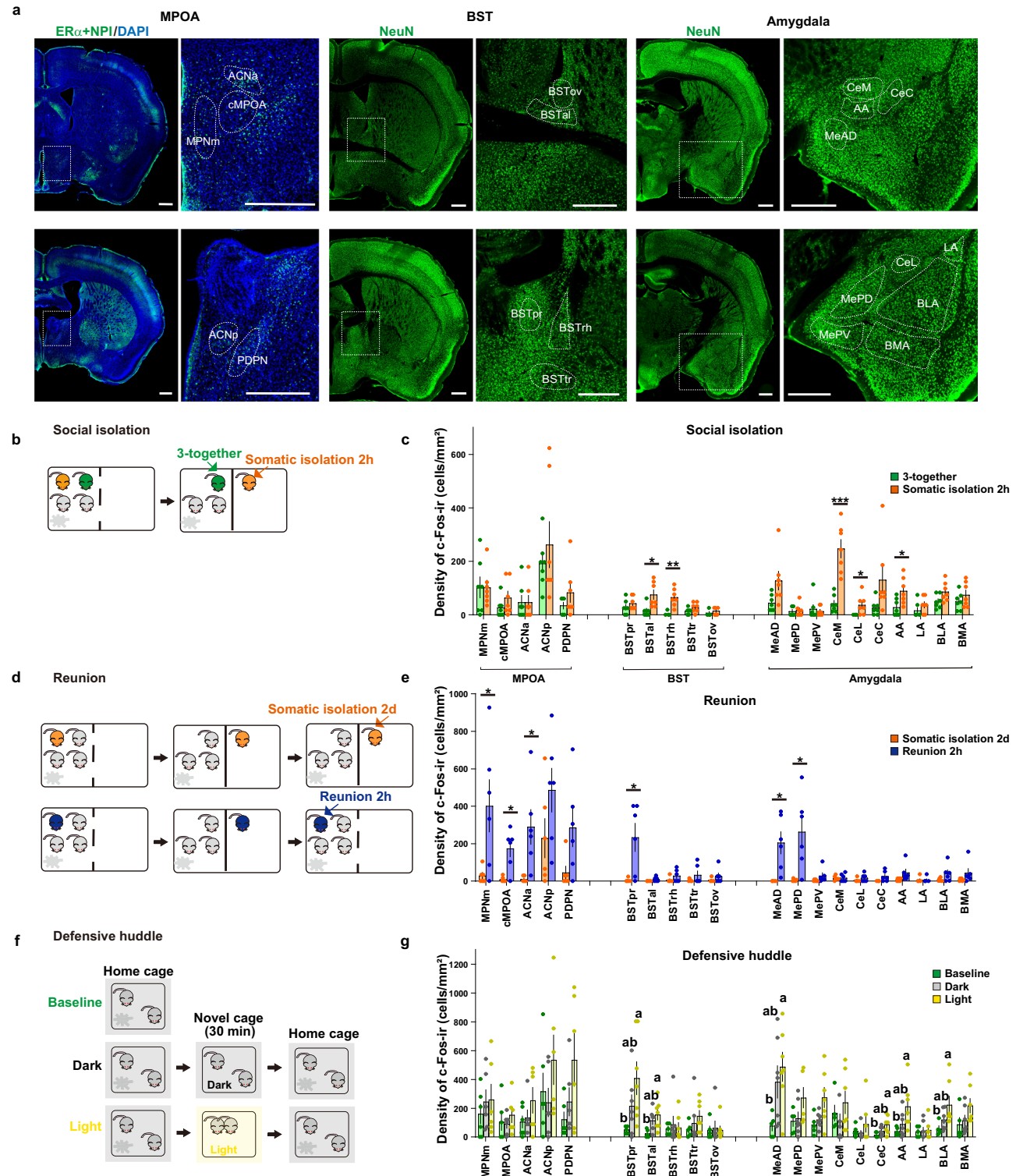

Chemogenetic activation of amylin+ neurons significantly increased the number of partition-biting behaviors during somatic isolation (Fig. 8c-e).

To ask if amylin is required for this behavior we used an *Amylin* gene-targeted (*Amylin*−/−) mouse line[83], which had been reported to be grossly healthy and fertile, with mild bone-related phenotypes[79]. After somatic isolation, *Amylin* −/− females exhibited a clear decrease of partition-biting behavior compared to *Amylin* +/+ females (Fig. 8f-h), with all the other behaviors intact. In contrast, *Amylin* −/− females did not show

alteration of reunion-induced social contact (Fig. 8i-k). Finally, partition biting after somatic isolation was also reduced by ovariectomy, and this reduction did not completely recover following estradiol supplement administration (Supplementary Fig. S8). Of note, we have previously reported that the *Amylin* mRNA expression is strongly dependent on estrogen and prolactin, and that estradiol supplement could only partially rescue amylin expression (Fig. 5 of[38]). Thus, the decrease of contact-seeking following ovariectomy may be attributed to the decrease in amylin expression in the cMPOA.

**Fig. 4 Activation mapping of the preoptic-strial-amygdala subregions after somatic isolation, reunion, and light-induced defensive huddle.**
**a** Conservative contours of the preoptic-strial-amygdala subregions, in which c-Fos-immunoreactive neurons were counted. Coronal sections were stained for NPI and ERα (both in green) and DAPI (blue) in the MPOA or NeuN (green) in the BST or amygdala. MPNm, the medial part of the medial preoptic nucleus; cMPOA, the central part of medial preoptic area; ACN, the anterior commissural nucleus; PDPN, the posterodorsal preoptic nucleus; BSTpr, the principal nucleus of the BST; BSTal, the anterolateral part of the BST; BSTrh, the rhomboid nucleus of the BST; BSTtr, the transverse nucleus of the BST; BSTov, the oval nucleus of the BST; MeAD, the anterodorsal part of the medial amygdala; MePD, the posterodorsal part of the medial amygdala; MePV, the posteroventral part of the medial amygdala; CeM, the medial part of the central amygdala; CeL, the lateral part of the central amygdala; CeC, the capsular part of the central amygdala; AA, the anterior amygdaloid area; LA, the lateral amygdala; BLA, the basolateral amygdala; BMA, the basomedial amygdala. Scale bars, 500 μm. **b, d, f** Somatic isolation (**b**), reunion (**d**), or defensive huddle (**f**) procedures. **c, e, g** Density of c-Fos-immunoreactive neurons (c-Fos-ir) in each subregion observed in controls and 2 h after the social condition change (3-together and 2 h of somatic isolation groups: $n = 7$ mice; somatic isolation 2d and 2 h reunion groups: $n = 6$; 2 h after the exposure to dark or light novel conditions: $n = 7$; baseline condition: n = 5). Asterisks in (**c**) and (**e**) indicate significant differences between 2 groups (Welch's unpaired $t$-test; *$p < 0.05$, **$p < 0.01$, ***$p < 0.001$). The letters indicate significant differences (Welch's ANOVA with Benjamini and Krieger FDR test, two-sided for data from the BSTpr and one-way ANOVA with Tukey's multiple comparison test for the other data, $p < 0.05$). Graphs show mean ± SEM. See Supplementary Data for exact $p$ values and details of statistical analyses.

Together, these finding argue that, among the various functions of Calcr+ neurons in facilitating social contacts, amylin plays a significant role in the induction of contact-seeking behaviors.

## Discussion

In this study we addressed the neuromolecular mechanism of affiliative contact behaviors among female mice, identifying a key role for amylin-Calcr signaling in the cMPOA. Calcr mediate behaviors required to form and maintain affiliative contacts with peers; that is, contact-seeking and exploratory behaviors immediately after somatic isolation, as well as actual contact after reunion (Fig. 6). This is similar to other hypothalamic-preoptic "centers" for goal-directed behaviors; for example, stimulation of the hypothalamic "drinking center" induces drinking behavior per se, as well as behaviors to remove an obstacle to gain access to water[84]; activation of Agrp neurons in the arcuate nucleus not only drives feeding when food is available, but also induces food-seeking behaviors when food is absent[85]. One distinction remains; while Agrp neurons are activated during hunger (lack of the goal), Calcr+ neurons are activated during reunion (achievement of the goal). To further elucidate the exact role of Calcr+ cMPOA neurons in social motivation, future studies should examine how Calcr+ neurons exhibit social condition-dependent activities with higher temporal resolution.

To our knowledge, even when compared to other molecules, such as Tac2, amylin expression in the cMPOA and ACN is the molecular marker most drastically regulated by social housing conditions[12]. And in turn, amylin influences social behaviors, as our experiment with *Amylin* KO demonstrated that amylin is required for contact-seeking behavior, but not for social contact per se (Fig. 8). Following both reunion and during continuous free social interaction we observed constitutive activity in amylin+ neurons, while social isolation led to deactivation (Fig. 1m-n). However these data raise an interesting paradox: amylin does not seem to be required when amylin neurons are activated, rather amylin seems to function when amylin neurons are *not* activated. However similar phenomena have been previously observed for GPCR-coupled neuromodulators; for example, retrograde endocannabinoid release can be driven by activation of various postsynaptic $G_{q/11}$-coupled receptors without need of postsynaptic Ca2+ elevation[86]. Dopamine release from terminals can also be locally controlled, and tonic dopamine release produces near-continuous stimulation of the D2 receptors, allowing the system to notice brief pauses of the dopamine cell firing as negative prediction errors[87]. Therefore, a tonic activation of amylin+ neurons (Fig. 1n) by social contact may maintain constitutive *Amylin* mRNA expression, which elicits immediate contact-seeking when the social contact is terminated by isolation. The gradual decrease of amylin expression by prolonged

social isolation might function as an adaptation to isolation; if the mice lack social contacts for a week, this would diminish needless contact-seeking behaviors. However, to validate such a working hypothesis, afferent connectivity and social condition-dependent activity-monitoring with high temporal resolution should be elucidated for amylin+ neurons. These data also indicate that, while amylin+ and Calcr+ neurons act in concert, Calcr+ neurons play the central role in the execution of social contact behaviors, with amylin+ neurons serving to modulate the function of Calcr+ neurons, similar to their respective roles in parental care[38].

It remains to be explained why amylin and its receptor Calcr-RAMP complex, a well-known regulatory system for appetite control[40,41], should also mediate affiliative social contacts and parental care. Of course, it is possible that amylin is used purely as a signaling component, and independently regulates food intake in the area postrema and social behavior in the cMPOA. Another possibility is that the level of amylin indeed conveys metabolic information along with social information to collectively regulate social behaviors. It has been known that the social organization of house mouse *Mus musculus* is highly variable, ranging from being solitary in non-commensal populations (i.e., inhabitant in fields, sand dunes), to forming high-density multimale/multifemale colonies in commensal populations (i.e., inhabitant in buildings or human settlements) with superabundant food supply[3]. Mice in a high-density commensal population are known to be less aggressive and more amicable[19]. Notably, amylin produced in the hypothalamus is reported to be scarce in males but are dramatically increased by high fat diet[88], thus it is reasonable to speculate that increased amylin by satiety could induce sociability in mice. On the contrary, food scarcity is one of the major limiting factors of group size, and individual animals increase solitary foraging to avoid intragroup food competition. In such cases, the limited amylin binding to Calcr in the cMPOA may downregulate contact-seeking behaviors to better adapt the environment. Further direct experimental demonstration is needed to prove this appealing possibility.

It should be acknowledged that the other factor(s) outside of the amylin-Calcr signaling may also play significant roles in facilitating social contacts, as the blockade of amylin-Calcr signaling exerts only partial effects on social contact behaviors. Oxytocin neurons have been demonstrated to be activated by social stimuli, and facilitate social preference and/or contacts in nonreproductive contexts[33,89–93]. Tang et al. have shown that somatosensation at the body-trunk by free social interaction, as well as by air-puffs, but not "chambered social interaction" (similar to our somatic isolation) activates oxytocin neurons in the PVH, and chemogenetic activation/inhibition of oxytocin neurons facilitate/inhibit social interactions among female rats, respectively. Because amylin+

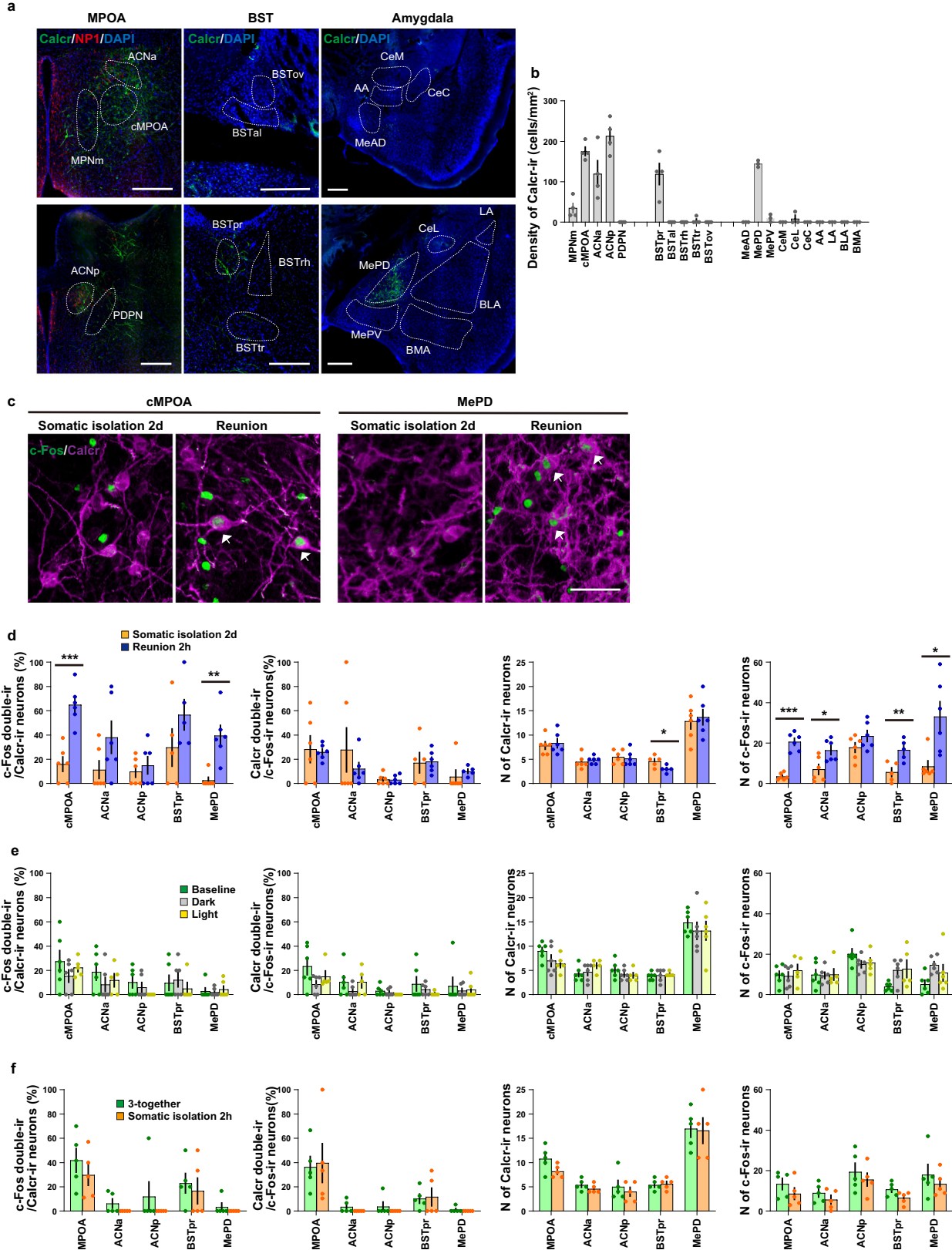

and Calcr+ neurons are distributed together with oxytocin neurons in the ACN, it is possible that amylin-Calcr neurons may act in concert with oxytocin neurons to sense and regulate affiliative social contacts. Very recent studies have reported that the MeA is transcriptionally activated by social reunion[94], and that GABA+ Tac1+ MeA neurons promote allogrooming toward a foot-shocked cage mate via their wide projections to the MPOA, but not to the ventral premammillary nucleus[95]. As MePD Calcr+ neurons are activated by social reunion (Fig. 5d), future work examining the relevant projections from the MeA to the MPOA in higher anatomical and molecular resolutions would be of interest to the field.

**Fig. 5 Distribution of Calcr+ neurons in the preoptic-strial-amygdala subregions and their activation during somatic isolation, reunion, and light-induced defensive huddle. a** Representative coronal brain sections showing the distribution of Calcr immunoreactivity (Calcr-ir, green), NPI-ir (red), and DAPI staining (blue) in the preoptic-strial-amygdala subnuclei. Scale bars, 300 μm. **b** Density of Calcr-ir neurons in each subregion. (MPOA and BST: $n = 4$ mice, amygdala: $n = 3$). **c, d** Distribution of Calcr-ir (magenta) and c-Fos-ir (green) neurons in the cMPOA or MePD after social reunion. Arrowheads indicate double-labeled cells. Scale bar, 50 μm. **d** Percentage of Calcr-ir neurons expressing c-Fos, percentage of c-Fos-ir neurons expressing Calcr, number (N) of Calcr-ir neurons and c-Fos-ir neurons in each subregion after reunion ($n = 6$ mice per group). **e** Percentage of Calcr-ir neurons expressing c-Fos, percentage of c-Fos-ir neurons expressing Calcr, number of Calcr-ir and c-Fos-ir neurons in each subregion after light-induced defensive huddle procedure (baseline and dark: $n = 6$ mice, light: $n = 5$). **f** Percentage of Calcr-ir neurons expressing c-Fos, percentage of c-Fos-ir neurons expressing Calcr, number of Calcr-ir and c-Fos-ir neurons induced in each subregion by 2 h of somatic isolation ($n = 5$ mice per group). Asterisks in (**d**) indicate significant differences between two groups (Welch's unpaired $t$-test, $*p < 0.05$, $**p < 0.01$, $***p < 0.001$). Graphs show mean ± SEM, dots represent individual data. See Supplementary Data for exact $p$ values and details of statistical analyses.

It has not been foreseen that the amylin-Calcr signaling in the cMPOA, which is identified for its critical role in parental care, also mediates affiliative contact behavior among females[38]. Retrospectively, however, it is in harmony with the long-postulated notion that the social bonding among adult animals evolved from parental care of offspring[4,96], by the fact that intricate affiliative relations among adults are found only in species that perform extensive parental care, such as mammals and Hymenoptera. Moreover, house mouse females engage in communal nursing, where the group of mothers giving birth at the same place and indiscriminately nurse all pups. Communal nursing is shown to be beneficial for pup growth[31], thus it is ethologically relevant if maternal motivation facilitates social housing via amylin-Calcr+ signaling in the cMPOA. The fact that amylin-Calcr system in the cMPOA is involved in both maternal care and social affiliation but not in defensive huddle have led us to distinguish two dimensions of social motivation; one is for survival or preservation of the physical self and enhanced by fear and environmental stress, including defensive huddle. The most primitive form of this type of social motivation is exemplified by "the selfish herd"[65]; "gregarious behavior is considered as a form of cover-seeking in which each animal tries to reduce its chance of being caught by a predator". Another dimension of social motivation is for reproduction, or preservation of the genetic self, and often involves at least some cooperation or affiliation with other conspecifics. The most primitive and extensive kind of such affiliative social behaviors is parental nurturing behavior, in which the caregivers even sacrifice their chance of survival for the sake of the care-receiver (generally offspring or kin in many group-living mammals). And the activation pattern by social reunion resembles more with maternal care than with defensive huddle. These two dimensions of social motivation have an interesting similarity with the two-dimensional aspects of human sociality and attachment[97-99], one being about the security of self and the other is about the interest toward others, prompting further attention.

Obviously, more work is required to elucidate the neural processes involving MPOA amylin-Calcr signaling to sense social isolation, seek for and engage in social contacts to enable complex mammalian affiliative behaviors. Such scientific knowledge should also contribute for better treatment and prevention of mental health issues caused by loneliness, which are widely appreciated under current pandemic conditions[100].

## Methods

**Animals**. All mouse experiments were approved by the RIKEN Animal Experiment Committee, complied with the ARRIVE guidelines[101], and were conducted in accordance with the National Institutes of Health guide (NIH publication no. 85–23, revised 1985).

BALB/c and C57BL/6 J mice were purchased from Charles River (BALB/c) and the Jackson Laboratory (C57BL/6 J) and were raised in our breeding colony. *Amylin* gene knockout (KO) mice (*B6.129P2-Iapp*[tm1Sgm/Kctt], RRID: IMSR_EM:05269) were provided by Samuel Gebre–Medhin via the European

Mouse Mutant Archive[83]. *Amylin*-KO and *Amylin*-Cre mice were backcrossed to BALB/c for at least five generations. The Ai3 strain *B6. Cg-Gt (ROSA)26Sor* [tm3(CAG-EYFP) Hze] */J* (JAX 007903) was from the Jackson Laboratory. Mice were maintained under controlled conditions (12/12 h light/dark cycle [lights on at 08.00 a.m.]; 23±2 °C; 55±10 % humidity with food and water ad libitum. Female mice were housed in groups of four or five after weaning at 4 weeks. Mice were 3-10 months old at the start of the experiments.

**Reagents**. The reagents used in the experiments and their commercial sources were as follows: Amylin (mouse, rat) trifluoroacetate salt (500 μg, BACHEM, Torrance, CA), ACSF (Tocris Bioscience, Bristol, UK), clozapine-N-oxide (CNO, Tocris Bioscience), and 2-methylthio-2-thiazoline (2MT, Tokyo Chemical Industry). CNO (10 mg/ml stock) was dissolved in saline (0.9 % NaCl) containing 20% (v/v) dimethyl sulfoxide (Wako, Tokyo, Japan). Amylin (mouse, rat) was dissolved in 150 μl ACSF.

### Behavioral experiments

*Housing status on Amylin mRNA expression.* The housing condition was manipulated to examine the effect of cage mate presence on amylin expression. For the experiment in the Fig. 1e, virgin female mice were group-housed (in groups of five) from weaning, and were subjected to perfusion fixation and brain dissection after 0, 2, 6, or 20 days of single-housing. Additional virgin females were single-housed for 6 days and then were group-housed back with their siblings. Samples from these mice were taken 2, 6, or 14 days after the second round of group housing. The effects of cage mates were investigated by housing virgin female mice with one or four unfamiliar females or with four castrated males. After 6 days of cohousing, their brains were analyzed.

To examine the effect of restraint stress on *Amylin* mRNA expression, group-housed virgin female mice were subjected to single-housing for 6 days. Then, they were assigned to the following groups: cohousing/stress + , cohousing/stress − , isolation/stress + , isolation/stress− for six additional days. The cohousing groups consisted of five mice per cage. The stress+ groups were subjected to restraint stress for 2 h daily for the 6 days using ventilated 50 ml plastic tubes as described previously[38]. At the end of the experiment, mice were subjected to perfusion fixation and the brains were dissected.

To examine the effect of the in-cage isolation on A*mylin* mRNA expression (Fig. 1p and Supplementary Fig. S1f), group-housed mice (in groups of five) were placed in a rat cage (23.5 × 45 × 20 cm) equipped with a small mesh cage (15 × 15 × 15 cm), purified paper bedding (Alpha-Dri), and two water bottles (Supplementary Fig. S1f). After 6 days of habituation, one of the five mice was placed into the mesh cage. Brain samples of in-cage isolated mice and one of the group-housed mice (in groups of four) were dissected after 6 days of social isolation.

*Social isolation and reunion test.* Group-housed mice (in groups of four) were placed in a chamber (44 × 24 × 15 cm) containing purified paper bedding (Alpha-Dri) and two water bottles (Fig. 2a) and divided into left and right compartments by a central removable partition. Mice had access to food pellets and water ad libitum in each compartment. Two types of partition were prepared: a sham partition, through which mice could freely go, and a real partition wall, through which mice could see and sniff each other but were not in free physical contact. After 1 week of habituation, one of the four mice was placed into the compartment without the previous nest, and the sham partition was replaced with the real partition. Their behavior was tracked for more than 3 h using a video camera (iVIS HF R52, Canon, Tokyo, Japan) and an interval camera (Recolo, KINGGIM, Tokyo, Japan) for recording.

Based on the previous ethogram[54,55], mouse behaviors were coded and categorized as follows: 1) *general movements*: still (no movement), movement (locomotion or any physical movements not otherwise specified); 2) *exploration of the environment*: sniffing (targeted at the partition or through the partition by putting the nose into the slit of the partition window), rearing, digging; 3) *contact-seeking*: biting (partition); 4) *nonsocial behaviors*: self-grooming, eating, panic.

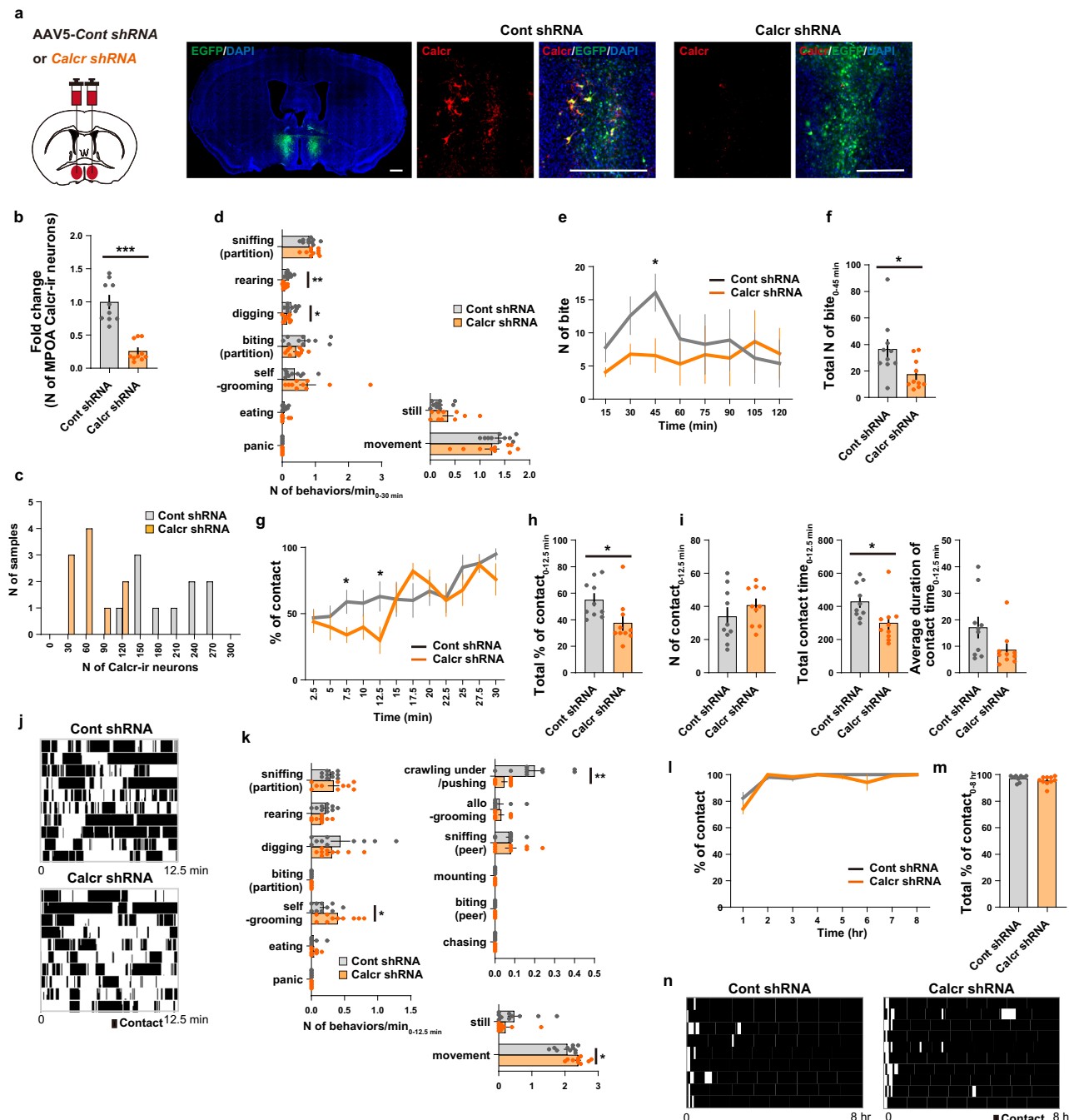

**Fig. 6 Targeted knockdown of *Calcr* in the cMPOA blocks behavioral responses to somatic isolation and reunion. a** Left panel: experimental procedure. Two weeks prior to testing, female mice were injected with AAVs expressing shRNA-EGFP for the specific knockdown of *Calcr* (Calcr shRNA) or with a virus carrying scrambled shRNA (Cont shRNA). Other panels: representative coronal sections of shRNA-EGFP viral expression in the cMPOA showing the efficacy of *Calcr* shRNAs. Scale bars, 500 μm. **b, c** Quantification of number (N) of Calcr-immunoreactive (Calcr-ir) neurons in the MPOA ($n = 10$ mice per group). The number of Calcr-ir neurons significantly decreased in the cMPOA after *Calcr* shRNA injection. **d** Quantification of behaviors (number of occurrences per min) seen during somatic isolation for 30 min ($n = 10$ mice per group). **e** Time course analysis of the biting responses ($n = 10$ mice per group). (**f**) Total number of biting responses occurring during 45 min ($n = 10$ mice per group). **g** Percentage of social contacts (number of occurrences divided by total number of observations at a time point) over time ($n = 10$ mice per group). **h** Percentage of social contacts (total number of occurrences divided by total number of observations) during 12.5 min. (**i**) Quantification of the number of contacts, total contact time, and average duration of contact time. These data were analyzed using continuous video recording during 12.5 min ($n = 10$ mice per group). **j** Raster plots showing the effects of *Calcr* shRNA injection in the MPOA on social contacts. **k** Quantification of behaviors (number of occurrences per min) seen during social reunion for 12.5 min ($n = 10$ mice per group). **l** Analysis of the percentage of social contacts (number of occurrences divided by total number of observations at a time point) over time (Cont shRNA: $n = 9$ mice and Calcr shRNA: $n = 10$). (**m**) Percentage of social contacts (total number of occurrences divided by total number of observations) during 8 h (Cont shRNA: $n = 9$ mice and Calcr shRNA: $n = 10$). **n** Raster plots showing the effects of *Calcr* shRNA injection in the MPOA on contacts. Time bins: 15 sec (**d-g, k**) or 5 min (**l-n**). Asterisks in (**b-f, g-i, k**) indicate significant differences between two groups (Welch's unpaired t-test, *$p < 0.05$, **$p < 0.01$, ***$p < 0.001$). Graphs show mean ± SEM. See Supplementary Data for exact $p$ values and details of statistical analyses.

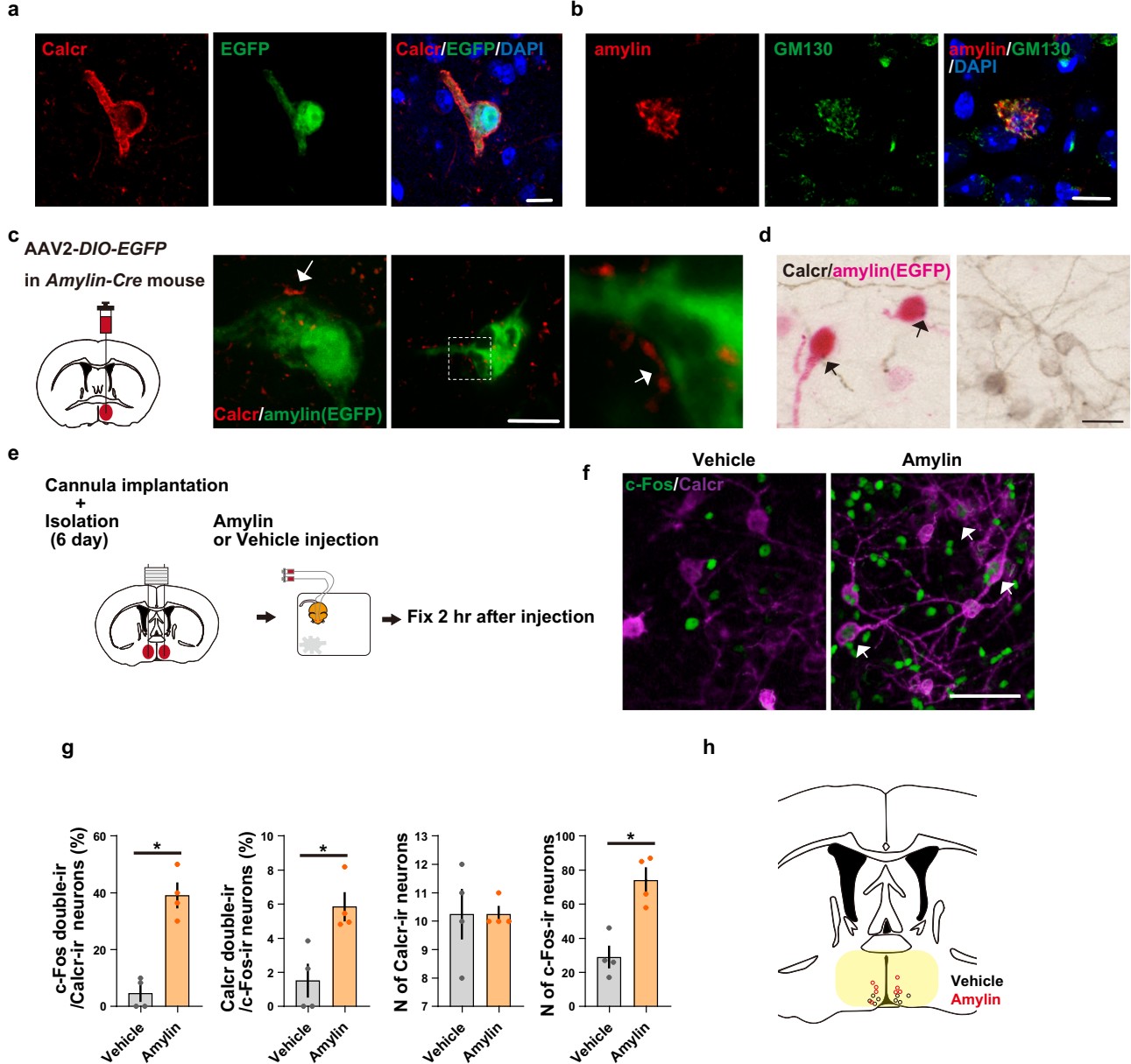

**Fig. 7 Spatial relationship between Calcr+ and amylin+ neurons and the activation of Calcr+ neurons by amylin.** (**a**) Localization of Calcr (red) on the plasma membrane of EGFP + neurons (green). Coronal sections of the cMPOA were stained with anti-Calcr antibodies and DAPI (blue). Scale bars, 10 μm. (**b**) Amylin localization in the Golgi apparatus. Sections were stained with antibodies recognizing amylin (red) and GM130 (green), a cis-Golgi matrix protein. Scale bars, 10 μm. (**c, d**) The anterograde fibers of Calcr-immunoreactive (Calcr-ir) neurons lie in the vicinity of amylin+ neurons (green). Coronal sections of the MPOA of group-housed *Amylin-Cre* female mice injected with Cre-dependent EGFP AAV vector into the cMPOA. Sections were stained with anti-Calcr (red or black) antibodies. Images were taken with confocal (**c**) or bright-field (**d**) microscopes. Scale bars, 10 μm (**c**) and 25 μm (**d**). (**e**) Experimental procedure. Six days before amylin infusion, group-housed female mice were equipped with guide cannula and single-housed. Two hours after local amylin or vehicle infusion in vivo, the mice were subjected to perfusion fixation for immunochemical studies. (**f**) Distribution of Calcr (magenta) and c-Fos (green) in the cMPOA with or without amylin infusion. Arrowheads indicate double-labeled cells. Scale bar, 50 μm. (**g**) Percentage of Calcr-ir neurons expressing c-Fos, percentage of c-Fos-ir neurons expressing Calcr, number (N) of Calcr-ir and c-Fos-ir neurons in the MPOA after amylin infusion ($n = 4$ mice per group). (**h**) Schematic coronal section showing the location of the injection cannula tips. Black (red) circles indicate individual cannula tips used for vehicle (amylin) infusion. The yellow zone corresponds to the assumed spreading area of the injected blue dye. Asterisks indicate significant differences between two groups (Mann-Whitney *U* test, two-sided, *$p < 0.05$). Graphs show mean ± SEM. See Supplementary Data for exact *p* values and details of statistical analyses.

Direct social behaviors were not coded as they cannot be displayed by isolated mice. The behaviors and the mouse position (in contact with the partition, in the half compartment near the partition, or in the half compartment away from the partition; see Fig. 2a) in the cage were scored using video recordings at 15-s intervals for 30 min. In addition, the biting behavior, i.e., the mice bite the window frames of the real partition with their head tilted, was scored in a one-zero fashion from still images extracted at 15-s intervals for 3 h (Fig. 2) or 2 h (Figs. 6, 8, S5, S8)

from video recordings. The hunched posture, in which mice tuck their feet beneath the body with their back hunched up, and crouch posture, in which mice tuck their feet beneath the body without their back hunched up, were scored in a one-zero fashion using still images extracted at 1-min intervals for 3-h.

Two days after somatic isolation, the real partition was replaced by the sham partition for reunion. The social contacts made by the formerly isolated mice with conspecifics were assessed using still images extracted by Batch Video to Image

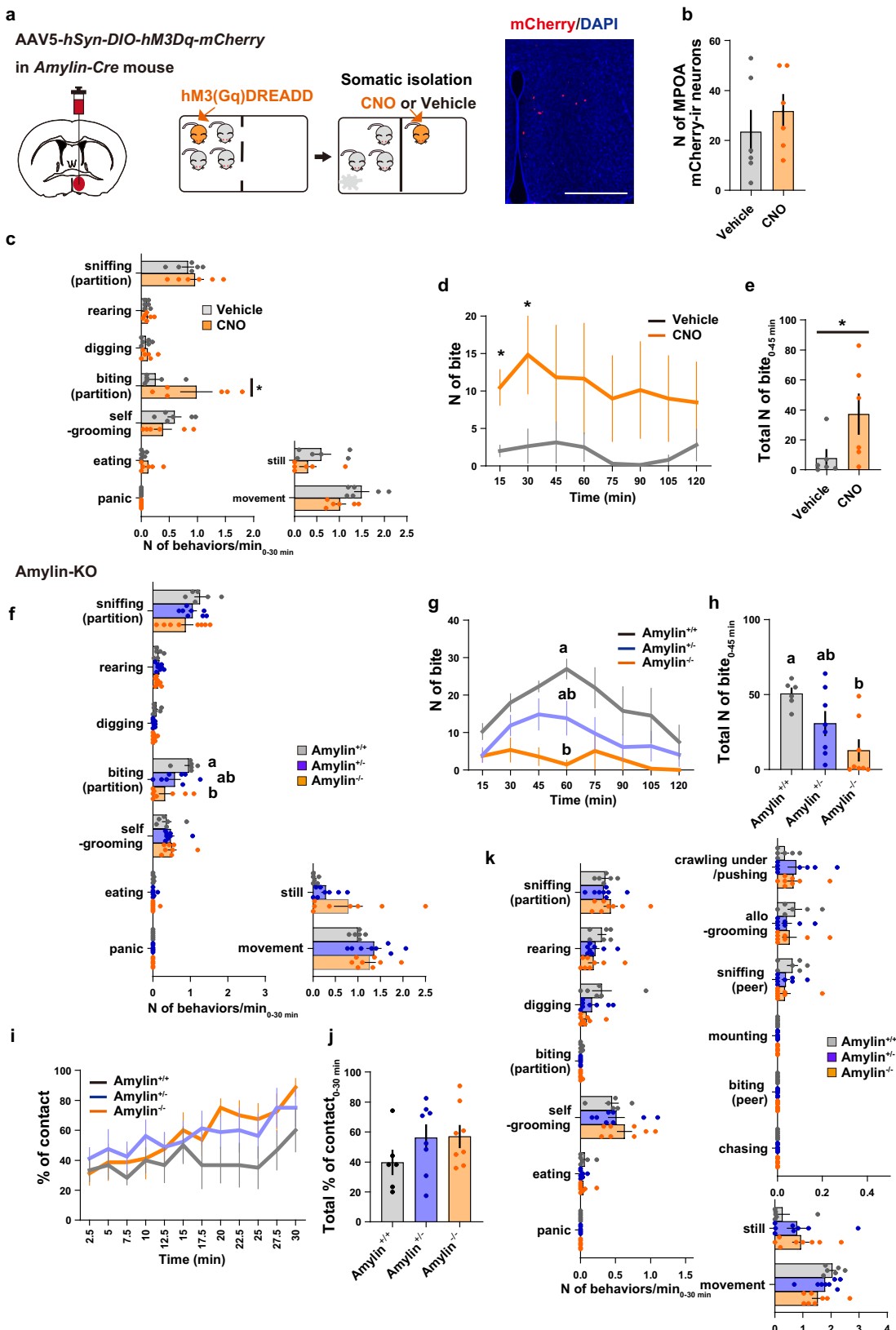

Extractor (version 0.1.7.) at 15-sec intervals during 60 min (Fig. 3c) or 30 min (Figs. 6, 8, S5, S8), and at 5-min intervals during 8 h using an interval camera. In addition to the above-described behaviors 1) ~4), 5) *direct social behaviors*, such as crawling-under/pushing, sniffing (peer), allogrooming, mounting, biting (peer), and chasing, were coded after reunion using video recordings. These behaviors were manually coded at 15-s intervals during 30 min. "Pushing" and "crawling-under" behaviors can be exhibited as an agonistic behavior by male rodents, but are

performed as a soliciting or amicable behavior by females[54,102]. The affiliative nature of these behaviors was confirmed by the lack of other kinds of agonistic behaviors in this study.

*Locomotor activity and core temperature measurement in social isolation or reunion test.* An activity measuring device, nanotag (Kissei Comtec Co., Nagano, Japan),

**Fig. 8 Amylin signaling is required for contact-seeking behavior. a–e** Contact-seeking and partition-biting behaviors were induced by chemogenetic activation of amylin+ neurons. *Amylin-Cre* female mice were unilaterally injected with AAV5-*hSyn-DIO-hM3D(Gq)-mCherry* in the MPOA. After 2 weeks, virus-injected mice were subcutaneously injected with CNO or vehicle 30 min before testing. **a** Experimental procedures and representative picture showing mCherry-ir neurons (red). Scale bar, 500 μm. **b** Number (N) of mCherry-ir neurons in the MPOA. **c** Quantification of behaviors (numbers of occurrences per min) observed after somatic isolation for 30 min. **d** Time course analysis of the biting responses. **e** Total numbers of biting responses during 45 min (n = 6 mice per group). Asterisks indicate significant differences between two groups (Mann–Whitney *U* test, *$p < 0.05$). **f–k** The somatic isolation-induced biting behavior was decreased in *Amylin* knockout female mice. **f** Quantification of behaviors (number of occurrences per min) seen after somatic isolation during 30 min. (**g**) Time course analysis of the biting behavior. **h** Number of biting responses observed after somatic isolation during 45 min. **i** Time course analysis of social contacts. (**j**) Percentage of social contacts (total number of occurrences divided by total number of observations) during 30 min. **k** Quantification of behaviors (number of occurrences per min) observed after social reunion during 30 min at 15-s intervals (amylin + /+: n = 6 mice, amylin + /- and amylin-/-: n = 8). The letters in (**f–h**) indicate significant differences (Kruskal–Wallis test with Dunn's multiple comparison test, two-sided, $p < 0.05$). Graphs show mean ± SEM. See Supplementary Data for exact *p* values and details of statistical analyses.

was implanted intraperitoneally to measure the mouse locomotor activity. The activity and core temperature were recorded at 1-min intervals.

*Defensive huddle test*. Group-housed mice (in groups of two) were brought to the behavioral testing room and allowed to acclimate to a 12/12 h light/dark cycle (lights on at 02.00 p.m.) for 3 days. On the testing day (Day 1), mice (in groups of two) were placed into a dark novel cage, their behavior was tracked for 30 min using an infrared camera (400-CAM035-2, Sanwa Direct) for video capture, and they were then returned to the home cage. The following day (Day 2), mice were placed into a dark or a brightly illuminated novel cage and their behavior was tracked for 30 min. To produce the bright light, a 93.2 ± 5.5 lux fluorescent lamp was placed above the novel cage. Mice were scored for whether or not they made social contacts with conspecifics for 10 min using CAPTIV-L2100 (TEA, France). The number of fecal boli deposited in the novel cage was quantified after each behavioral test as anxiety measurement. Behavioral tests were performed during the dark period (10.00 a.m.–02.00 p.m.). The predator odor-induced defensive huddle test was conducted in the home cage. Group-housed C57BL/6 female mice (in groups of two) were acclimate to a 12/12 h light/dark cycle (lights on at 08.00 a.m.) for 3 days. On the testing day, mice (group of two) were brought to the behavioral testing room equipped with a draft chamber and allowed to acclimate for 30 min. A piece of filter paper (2 × 2 cm) soaked with 2MT (270.6 μmol) was placed at the edge of the home cage and the mouse behavior was tracked for 30 min.

**Generation of the *Amylin-Cre* transgenic mouse line**. The bacterial artificial chromosome (BAC) modification was carried out using BAC recombineering technology. The BAC clone RP23-438K19, including the *Amylin* promoter, and gene, was transferred to DH10α E. coli cells. The *organic anion transporting polypeptide 3, Solute carrier organic anion transporter family member 1A5, ATP-dependent DNA helicase Q1*, and *pyridine nucleotide-disulphide oxidoreductase domain 1* genes were included upstream or downstream of *Amylin* gene in the RP23-438K19 clone. *NotI* sites were inserted 5′ and 3′ of *Amylin* gene using the Red/ET Recombination System to cut off superfluous genes. The *Cre* sequence was inserted in-frame into the second to third exons shared by the major *Amylin* isoforms (NM_010491). The *Cre* insertion cassette consisted of a 5′ homology arm (5′-GAAAGAGGTAGCTTGATTTCTGTGGGTTTTTTTTTTTTCTTTTCAGGG ATCTTGAGAA-3′), the Cre coding region, the *Ampicillin*-resistant gene flanked by FRTs, and a 5′ homology arm (5′-TCAGGAAATCACCCAGAGCATTTACA CATAAAGTATTAAGTGCTGCAGAAGTACATTGACT-3′). The purified BAC modifying cassette was electroporated into DH10α cells containing the RP23-438K19 BAC and screened by polymerase chain reaction (PCR). The ampicillin-resistant gene was removed by the FLP/FLPe recombinase encoded by the 706-Flp plasmid. Fingerprinting analysis using BamHI restriction enzyme and DNA sequencing were carried out to confirm the presence of the correct modifications in the BAC clone. The purified BAC DNA, including the transgene, was linearized by *NotI* restriction enzyme, and the vector backbone was removed. The linearized BAC DNA was purified with Sepharose® CL-4B (Sigma-Aldrich, St. Louis, MO). The purified BAC insert was microinjected into fertilized eggs (C57BL/6 J) at the pronuclear stage. Successfully developed two cell-stage embryos were transferred to pseudopregnant recipient Jcl:ICR females (Clea Japan). We identified one mouse line that exhibit faithful *Amylin-Cre* expression in neurons expressing endogenous *Amylin*. This *Amylin-Cre* mouse line was then confirmed to be normal for the growth, general health, and various behaviors, including general home-cage behaviors, mating, nest building, virgin female nurturing behavior, postpartum maternal behaviors, and activities on the elevated plus maze.

Transgenic mice were genotyped by PCR using three primers (5′-CTTTGATG GTTCCAATTTACACT-3′, 5′-AAGTAGAAATCTTCAGGTCA-3′, and 5′-TCCG GTTATTCAACTTGCACCATGC-3′) to detect a 205-bp sequence present in the *Amylin-Cre* transgene and a 303-bp sequence present in the *Amylin* gene.

**Viral constructs**. AAV2-*hSyn-DIO-EGFP* (AV4501B) was purchased from UNC GTC Vector Core. To construct small hairpin RNA (shRNA), we used pAAV-*hH1-*

*MCS-CAG-EGFP* plasmid donated by M. Kengaku. The web-based software siDirect was used to design shRNAs for the mouse Calcr gene (BC_119272). Oligonucleotides containing the target sequence, loop region (tgtgctt), and anti-sense sequence were inserted into the multi-cloning site of pAAV-*hH1-MCS-CAG-EGFP*. The target sequences are as follows: *Calcr* shRNA: 5′-GGATCTATCTT CATACTCTGA and scrambled shRNA (control shRNA): 5′-GGTTTCTAGAC TCCTACTAAT. Adeno-associated viruses (AAVs, 1.0E11–1.0E12 viral genome [vg]/ml) were made by Thomas McHugh's laboratory as described previously[38]. 293FT cell line (Invitrogen) was used in AAV production.

**Histochemistry**. Mice were anesthetized with xylazine hydrochloride (0.24 mg, i. p.; Bayer Yakuhin, Ltd, Osaka, Japan) and lethal amount of sodium pentobarbital (3.24 mg, i. p.; Kyoritsu Seiyaku, Tokyo, Japan) and then perfused with 4% PFA in phosphate-buffered saline (PBS). The brains were immersed in 4% PFA at 4 °C overnight and then in 30% sucrose in PBS for 2 days. They were embedded in O.C.T. Compound (Tissue-Tek, Sakura Finetek, Tokyo, Japan) and cut into 40-μm coronal sections using a Cryostat (Leica CM1950). Every third brain section was prepared for histochemistry.

For c-Fos, Calcr, and amylin mapping experiments, brain sections were washed twice with PBS containing 0.2% Triton X-100 (PBST) for 10 min, treated with methanol containing 0.3% $H_2O_2$ for 5 min, washed three times with PBST for 5 min, and treated with 0.8% Block Ace (Megmilk Snow Brand Co., Ltd, Tokyo, Japan) in PBST for 30 min for blocking. Sections were then incubated with anti-c-Fos (1:5,000, sc-52, Santa Cruz Biotechnology, Dallas, TX, USA), anti-Calcr (1:4,000, PAb188/10, Welcome receptor antibodies, North Melbourne, Australia), or anti-amylin (1:20,000, H-017-11, Phoenix Pharmaceuticals, Burlingame, CA, USA) antibodies diluted into 0.8% Block Ace PBST overnight at 4 °C. Afterward, sections were washed and incubated with biotin-conjugated horse anti-rabbit secondary antibody (1:2000, BA-1100, Vector Laboratories, Burlingame, CA, USA) for 2 h and then with ABC peroxidase reagent (Vector Laboratories). Signal-positive cells were detected by diaminobenzidine (DAB) reaction (Vector Laboratories). For double staining, the sections were similarly incubated with an anti-NeuN (1:5000, MAB377, Merck Millipore, Temecula, USA) or anti-ERα (1:20,000, 06-935, Merck Millipore) antibody in combination with an anti-Neurophysin I (NPI) antibody (1:2,000, sc-7810, Santa Cruz Biotechnology) and then with an anti-mouse (1:2000, BA-2000, Vector Laboratories) or anti-rabbit secondary antibody in combination with an anti-goat secondary antibody (1:2000, BA-9500, Vector Laboratories). The sections were then incubated with the ABC alkaline phosphatase reagent (Vector Laboratories). Signal-positive cells were detected with ImmPACT Vector Red (Vector Laboratories).

For double labeling of Calcr and c-Fos or amylin and c-Fos, c-Fos-immunoreactive neurons were detected by DAB reaction with nickel. Following the first staining, Calcr- or amylin+ cells were detected by DAB reaction without nickel.

For immunofluorescence, brain sections were washed three times with PBST for 10 min and treated with 0.8% Block Ace in PBST for 30 min for blocking. Sections were then incubated with the primary antibody diluted into 0.8% Block Ace PBST overnight at 4 °C. The next day, sections were washed, incubated with the secondary antibody, washed once with PBST for 5 min, incubated with DAPI diluted in PBST for 5 min, and washed with PBST for 5 min. Finally, the sections were mounted between slide and coverslip using Vectashield Vibrance antifade mounting media (Vector laboratories). The signal of some primary antibodies was enhanced by biotin-Streptavidin-Alexa Fluor 568 (Invitrogen). We used the following primary and secondary antibodies: anti-NeuN (1:5,000, MAB377, Merck Millipore), anti-ERα (1:20,000, 06-935, Merck Millipore), anti-NPI (1:2,000, sc-7810, Santa Cruz Biotechnology), anti-Calcr (1:4,000, PAb188/10, Welcome receptor antibodies), anti-amylin (1:20,000, H-017-11, Phoenix Pharmaceuticals), anti-GM130 (1:1000, 610822, BD Biosciences), and Alexa 488- or Alexa 568-conjugated anti-rabbit, anti-mouse, or anti-goat IgGs (1:1,000, Invitrogen, or Abcam).

Single in situ hybridization (ISH) of *Amylin* mRNA was performed as previously described[37] with modifications. To make ISH probes, the nucleotides 70–577 of *Amylin* (NM_010491) were amplified by PCR from mouse brain cDNA

(GenoStaff) and an SP6 RNA polymerase recognition site was added to the 5′ end of the PCR product using specific primers (sense: 5′-CCTCGGACCACTGAAAG-3′; antisense: 5′-ATTTAGGTGACACTATAGAAACATTGACTTCACT CTGAACTTGATCA-3′). The antisense probes were transcribed by SP6 RNA polymerase (P2083, Promega) in the presence of DIG-RNA labeling mix (Roche Diagnostics, Basel, Switzerland) and isolated by lithium chloride precipitation.

Brain sections were washed twice with PBST for 5 min, treated with PBST containing 20 μg/ml Proteinase K (Invitrogen) for 10 min, treated with 4% PFA in PBS for 10 min for post-fixation, incubated with methanol containing 0.3% $H_2O_2$ for 5 min, and washed twice with PBST for 5 min. Then, they were treated with 0.25% acetic anhydride in 0.1 M triethanolamine and washed twice with PBST for 5 min. Brain sections were treated with prehybridization solution at 58°C for 30 min and hybridized at 58°C overnight in the probe containing hybridization solution. After hybridization, the sections were incubated with 2x SSC containing 50 % formamide twice at 58°C for 10 min, with RNase A solution (20 μg/ml) at 37°C for 45 min, with 2x SSC, and finally with SSC 0.2x SSC twice at 37°C for 10 min. They were washed with TBST for 5 min and incubated with an alkaline phosphatase-conjugated anti-DIG-AP (1:10000, Roche) antibody at 4°C overnight. Afterward, sections were washed twice with TBST for 5 min and incubated in 100 mM tris-HCl (pH 9.5) for 5 min. Signal-positive cells were detected with the BCIP/NBT solution kit (Nacalai Tesque, Kyoto, Japan).

For ISH combined with Calcr IHC, mRNA of VGLUT2 (2427-3006 bp, NM_080853.3) or GAD67 (1064-2045 bp, NM_008077) were detected by ISH. These ISH probes were used in the Allen Mouse Brain atlas. Following the ISH, Calcr+ cells were detected by DAB reaction without nickel.

**Image acquisition and analysis**. Sections were imaged by NanoZoomer Digital Pathology (Hamamatsu Photonics, Shizuoka, Japan) with a 20x or 40x objective for bright-field observation and BZ-9000 All-in-one Fluorescence Microscope (Keyence, Osaka, Japan) with a 20x or 40x objective for fluorescence observation. Confocal images were acquired using a confocal microscope with a 40x objective (FV1000, Olympus, Tokyo, Japan).

For c-Fos, Calcr, and amylin mapping, subregions in the preoptic area, BST, and amygdala were determined using the brain atlas[103] and previous publications[36,37]. The conservative contours (Fig. 4a) were set on brain sections based on counterstaining of ERα and NPI or NeuN. To quantify the number of c-Fos or Calcr-ir cells, acquired images were binarized and labeled neurons within each conservative contour were automatically or manually counted using ImageJ (NIH). For data shown as densities (N of cells/mm²), the number of labeled neurons was divided by the area of each conservative contour calculated using ImageJ.

**Stereotactic injection**. Stereotactic coordinates for AAV injections were taken from the mouse brain atlas[103]. The coordinates for the cMPOA were AP + 0.1, ML ± 0.55, DV − 5.1. Virgin female mice were anesthetized with xylazine hydrochloride (0.24 mg, i. p.; Bayer Yakuhin, Ltd) and ketamine hydrochloride (2.4 mg, i. p.; Daiichi Sankyo, Tokyo, Japan) and mounted in a stereotaxic apparatus (Narishige, Tokyo, Japan). A volume of 100 nl AAVs was injected into the targeted brain region through a pulled fine glass capillary (20–50 μm diameter at the tip) by oil pressure at a rate of 25–50 nl/min. Capillaries remained in place for 5 min allowing the solution diffusion. After surgery, the incision was sutured, and the mice were warmed-up until recovery from anesthesia.

**In vivo chemogenetic activation mediated by DREADD-Gq**. Subject virgin female Amylin-Cre mice were raised in groups of five or four until they were more than 3 months of age. A total of 100 nl of AAV5-hSyn-DIO-hM3D(Gq)-IRES-mCherry (5.6E12 vg/ml, AV44951, UNC Vector Core, Chapel Hill, NC, USA) was unilaterally injected into the cMPOA at an approximate speed of 50 nl/min. Afterward, each virus-injected Amylin-Cre mouse was transferred into a test cage and cohoused with three wild-type virgin female mice. After 1 week of habituation (2 weeks after AAV injection), the mice were subjected to behavioral testing. Stocked CNO was dissolved in saline and injected subcutaneously at 10 mg/kg for Gq-DREADD activation prior to somatic isolation or reunion test. Control injections were made with 2% DMSO in saline solution. The virus-injected mice (one in four mice) were socially isolated in the nestless part of the test cage and biting responses were observed for more than 2 h. The expression of mCherry was retrospectively examined using an anti-mCherry antibody (1:5000, ab167453, Abcam).

**Local amylin infusion**. For amylin infusions, guide cannula (C235G-1.0/SPC, Plastics one, Roanoke, UA, USA) were placed stereotaxically and bilaterally into the MPOA (AP, + 0.05 mm; ML, + 0.5 mm; DV, -4.6 mm). Injector cannula (C235I-1.0/SPC, Plastics one) protruded 1.0 mm from the tips of the guide cannula. Canula-implanted mice were housed individually for 6 days. For microinfusion of amylin, the injector cannula was attached to Hamilton syringes (26201, 5 μl) with connector assemblies (C232CS, Plastic one). The bilateral infusion of amylin (300 nl ACSF with 1 μg amylin at a speed of 50 nl/min) was controlled by a microinfusion pump (ESP-36, Eicom, Kyoto, Japan). The brains were fixed and

subjected to immunochemical studies 2 h after infusion. The correct placement of the injector cannula tip was confirmed in brain sections.

**N-methyl-D-aspartic acid (NMDA) Lesion**. Virgin female mice were raised in groups of five or four until they were more than 3 months of age. A total of 40 or 100 nl of NMDA (Sigma-Aldrich) and NMLA, Santa Cruz Biotechnology) dissolved in PBS at 20 mg/ml was injected bilaterally into the cMPOA or the whole posterior MPOA including the medial part, respectively. The coordinates were AP: + 0.1, ML: ± 0.4-0.55, DV: -5.1 to -5.4 for the MPOA and AP: + 0.1, ML: ± 0.55, DV: -5.1 for the cMPOA. A week after injection, behavioral tests followed by immunochemical studies were performed. The lesion location was identified by immunostaining with a mouse anti-NeuN antibody (1:5000, MAB377, Merck Millipore).

**Ovariectomy and estrogen treatment**. Virgin female mice were anesthetized with xylazine hydrochloride (0.24 mg, i. p.; Bayer Yakuhin, Ltd) and ketamine hydrochloride (2.4 mg, i. p.; Daiichi Sankyo, Tokyo, Japan) and the ovaries were removed (ovariectomy, OVX). Then, the incision was sutured, and the mice were kept warmed until recovery from anesthesia. Sham controls underwent the same procedure except that the ovaries were not removed. Each operated mouse was cohoused with three intact virgin females in a test cage. After one week of habituation, social isolation and reunion tests were performed. Estrogen was administered using an osmotic pump as described previously[104]. Briefly, ovariectomized mice were treated with 17beta-estradiol (E2, Sigma-Aldrich, St. Louis, MO) (2 μg/day) via subcutaneously implanted Alzet osmotic pumps (Alzet, DURECT Corporation, Cupertino, CA; model 1002; 14-day capacity at 0.25 μl/h infusion rate). E2 was dissolved in USP-propylene glycol (Sigma-Aldrich), which was used as the vehicle. Pump implantation was performed soon after the ovariectomy.

**Statistics and reproducibility**. Data were analyzed by unpaired t-test with Excel or nonparametric test with the Prism8 software for single comparisons. For multiple comparisons, analysis of variance (ANOVA) with post hoc tests were performed with the software R (R Development Core Team, 2019) or Prism8 (GraphPad Software, San Diego, CA, USA). The biological replicate number corresponds to the number of mice. Technical replicates were not applicable or performed in this study. All fluorescent image analyses were repeated independently at least two times, found to show a similar trend.

**Reporting summary**. Further information on research design is available in the Nature Research Reporting Summary linked to this article.

## Data availability
NA. Source data are provided with this paper.

## Material availability
The materials generated in this study are available from the corresponding author upon reasonable request with a completed Materials Transfer Agreement. Source data are provided with this paper.

## Code availability
This study did not generate any unique code.

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

## Acknowledgements

We wish to thank Michael Numan for professional and insightful comments, Yousuke Tsuneoka for supporting the work of T.M., Yukimi Arai, Eri Miyazawa, Hazuki Inoue and Hiroka Matsubara for technical assistance, Mineko Kengaku for the pAAV-hH1-MCS-CAG-EGFP plasmid, R. Jude Samulski and the Vector Core at the University of North Carolina at Chapel Hill for the AAV2-hSyn-DIO-EGFP, Samuel Gebre-Medhin and the European Mouse Mutant Archive for the *Amylin* KO mouse strain, and the RIKEN Research Resource Division for BAC microinjection to create *Amylin-Cre* mice and maintenance of the animals. This research was supported by RIKEN Center for Brain Science (2012-2021, KK), RIKEN Special Post-Doctoral Research fellowship (202001061045, K.F.), JSPS PD (2017-2020, K.F.), JSPS KAKENHI Grant Number JP16K19011 (K.F.) and partly by 18H02710 and AMED under grant JP20dm0107144 (K.K.).

## Author contributions

K.F. designed, carried out experiments and wrote the manuscript with K.K. M.K. carried out experiments for Fig. 1f-j, o-q, S1 under supervision of C.Y., K.K., and M.T., and assisted the works of K.F. T.M. carried out experiments for Fig. 1c-e under supervision of K.K. and M.T. C.Y. constructed the *Amylin-Cre* Tg mouse line under supervision of SI, carried out experiments for Fig. S7, histological analyses for Fig. S7. AJH prepared viral solution used for Figs. 6, 7, and S7 under supervision of T.J.M. K.O.K. designed and organized the study, and wrote the manuscript with K.F., T.J.M., and with contributions from all the authors.

## Competing interests

The authors declare no competing interests.

## Additional information

**Peer review information** *Nature Communications* thanks Stefanos Stagkourakis and the other anonymous reviewer(s) for their contribution to the peer review this work. Peer reviewer reports are available.

