## [Peer Review File · Nature Communications]

Reviewers' Comments:

Reviewer #1:

Remarks to the Author:

The present study by Fukumitsu et al. is a unique and important investigation on how the calcitonin receptor signaling in the MPOA mediates affiliative social behavior in adult female mice. They recognize the relevance of performing most of these experiments in female mice, escaping the restrictions of studying affiliative social behavior in males (due to intermale aggression). They use robust behavioral paradigms, and through clear quantifications, they suggest the significance of Amylin mRNA expression in social behavior. I have no doubt that this manuscript will be a significant addition to the literature.

It is important to note, however, that this manuscript requires attention from the authors and a major revision. Despite the wealth of data, I often found the writing and the flow of concepts in the text and figures confusing. Below I provide an outline of the points I would like to see clarified, with the aim that future readers can appreciate better the content of this work.

Major points

1. The authors make a case that the use of females is a better option for the study of affiliative social behaviors. They make a case for this in the Introduction (lines 64-85), but I would like to see this rationale reinforced.

The question "Why study this on females?", needs to be explicitly clarified to the reader. The authors also cite literature on maternal care – which I understand that they do because of the other study they have on review with similar findings (citation 47). In the introduction, however, I found such text and references distracting. The point of the present manuscript is to present an investigation on the neuro-molecular mechanisms that drive affiliative behaviors among adult mice. I would suggest only slight deviations for this line, by citing literature in maternal care etc.

2. Have the authors considered the effects of the estrus cycle in their behavior readout? This is important. What are the behavioral responses of ovariectomized mice in the social isolation test (presented in Figure 2)? What are the behavioral responses of ovariectomized mice supplemented with chronic estrogen-releasing mini-pumps in the same test?

3. How do the authors explain that the Amylin peptide – known for its role in energy metabolism and anorexigenic effect, has a role also in social behavior?

What is their understanding of the interplay between energy metabolism, and social behavior, through what is seemingly common mechanistic substrates?

Are they studying the effect of energy metabolism-related changes on behavior, or is Amylin a common substrate for both food intake AND social behavior? The reader would benefit greatly from understanding this, and the inclusion of such text would be relevant in the Introduction and/or Discussion.

4. I observed a general tendency by the authors to subjective/anthropomorphic interpretation of the data across the manuscript. While I cannot provide an exhaustive list of this, here is one example.

line261: "female mice gradually calmed down"

I urge the authors to report quantification of behavioral parameters, and not include writing indicative of a subjective interpretation of the data - especially in the Results section.

5. A methodological point: regarding the use of a mesh for partitioning cage compartments, I suggest to the authors the use of a plastic (PVC or equivalent) mesh instead. Mice will naturally exhibit biting behavior towards a metal mesh with round wires, especially upon the initial introduction of such a mesh to their cage – when the mesh is a novel object. Have the authors habituated their mice to this material prior to its use in the experiments of Figure 2? Could this affect the behavioral readouts presented in Figure 2?

6. This study uses a number of uncommon terms that need to be explicitly defined. I strongly recommend to the authors that when they use such terms, they use writing similar to the following rephrased example (relevant to lines 260-263):

"We often observed that mice, following reunion, formed a cluster and remained inactive or asleep,

a behavior we defined as sleep huddle. Sleep huddle has an obvious role in thermoregulation..."
Such clear definitions should be provided in the main text for:
a) defensive huddle, b) protest behavior, c) despair behavior, d) full social contact, e) somatic isolation

7. Throughout the Figures, the authors place the Y axis label in all graphs, in the place where one commonly places the title of the graph. Place the Y axis label, next to the Y axis in all figures.

8. If I understood correctly, the authors use C57 mice for the data presented in Figure 1, and BALB/c for the data presented in all other figures. I suggest that the authors move Figure 1 to the Supplementary material, and repeat the experiments presented in Figure 1c-e, and Figure 1p, q in BALB/c mice. All main figures, should be presenting data from the same mouse strain.

9. Perhaps I missed the point of this, but to my understanding Figure 9 does not belong to this manuscript. I found it irrelevant and superfluous, and therefore suggest its exclusion from the revised version.

Minor comments

Relevant to Major Point #6. In the Introduction, the authors cite nomenclature I am unfamiliar with, such as "protest behavior" and "despair phase". The sentences containing these concepts have no citations. Are the authors defining new concepts at this point in the manuscript, or are they using these concepts from previous literature. Please clarify.

Figure 1p: change samatic isolation to somatic isolation

line 46: change "sociable animals" to "social animals". Consider performing this change throughout the manuscript.

line 48: change "mood" to "behaviors". Identifying the mood of mice is difficult, but tracking their behavior is not.

line 52: change "(for example, 11)" to "(for example, Zelikowsky et al.11)

line 55: correct the notation of citations (" 12-18 19 ")

line 62-63: check whether this sentence "However, it remains poorly investigated if 63 isolation induces such a series of acute reactions in adult mice." Should be rephrased as "However, it remains poorly investigated if 63 isolation induces such a series of acute reactions in mouse pups."

line 73: change "sociality" to "sociability"

line 80, 81: "With such ethological considerations, it is not surprising that male mice may not exhibit depression by social isolation but rather show stress responses to group housing 44 25 45."
Reviewer: I do not agree with this sentence, single housing is stressful for male mice, as the author note on line 52 and by citing Zelikowsky et al.
This sentence and concept require revision.

Line 121: "A two-way ANOVA revealed...". Consider rephrasing this sentence. A statistical test permorms a comparison between two data sets. The experiment revealed your finding.

Line 127: The authors discuss in the text that environmental enrichment did not induce higher mRNA expression in isolated females, but they don't show the data. I recommend that this data set is included in the revised version as a supplementary figure.

Line 136: Ambiguous phrasing. What is a "full social contact"? Following this line, the authors define what they mean by somatic isolation, and the reader can infer what a full social contact

means. However if they want to use this phrasing, they have to define what a full social contact is prior to using the term in the text.

Line 144: change "could not able to" to "did not"

Lines 146-149: rewrite sentence, hard to read/understand

Lines 153-155: "The isolated female, however, seemingly perceived social isolation; soon after containment in the compartment, the female vigorously bit, dug beneath, and climbed up the wall, apparently trying to reunite with conspecifics."

Here the authors suggest the intent of the mouse. That is, that it bit the mesh, dug beneath, and climbed up the wall to reunite with conspecifics. Scientific experiments and writing should not allow for such subjective interpretations over why a mouse behaves a certain way. The field of systems neuroscience uses behavioral tests, and studies report behavioral quantifications when using these tests. Writing of this type does not belong to scientific articles, not in the Results, and neither in the Discussion.

Lines 227-229: "separated BALB/c adult female mice exhibit a two-stage separation distress response consisting of the first active "protest" phase of about 45 min, followed by the passive "depressive / despair" phase". These phases were presented in the introduction as behavioral phases of pups or infants. Here the authors use the same terminology in adult mouse behaviors.

Authors often write in the legends (n = # per group). Please specify units. Do you refer to mice, sections, etc.?

Figure 1a. Present the images at larger size in the figure.

Figure 1b. Are there any significant differences between brain areas? Illustrate this in the graph. In the legend of Figure 1b, it's written (n=5). 5 mice, 5 sections per mice? Please clarify.

Figure 1d. Increase the size of the images. Increase the contrast of the images, so as the immunoreactive cells become more visible.

Figure 1g. do the same as suggested for Figure 1d.

Figure 2m: Increase the size of the images. Increase the contrast of the images, so as the immunoreactive cells become more visible.

Reviewer #2:

Remarks to the Author:

In this study, Fukumitsu et al. have characterized the behavioural sequence of isolation and reintroduction of company in female mice, as well as its correlates in brain activation, with a focus on the amylin-calcitonin receptor system. Amylin neurons in the MPOA are shown to be sensitive to social conditions, as amylin expression is strongly depleted by isolation, and recovered by reunion with peers. Attenuation of signalling in this pathway by gene knockdown or pharmacological antagonism diminishes the amount of interaction during reunion.

This is an ambitious study on an intriguing topic; there is much work currently being published on the brain substrates for social behaviour, but the hypothalamic amylin system has received limited prior attention. Substantial effort is here made to work out the protocol for social isolation, which I would expect will be adopted by other investigators in the future. The authors have been careful to include appropriate controls, to determine which aspects of contact with peers cause changes in neuronal activation. Taken separately, each (or most) of the experiments are technically solid, and the level of detail and insightful description of methods is likely to be of great value to many others in the field as they design their experiments. (There are some drawbacks, however: e.g. the use of NMDA lesions is inferior in precision for targeting a particular population compared with genetically based approaches.) This is a great strength of the work.

The weakness of this study is that the data from the individual experiments are not coherently connected and the proposed amylin system is based largely on correlations. As some examples, it is not clear what the significance is of the demonstration that (lines 328-9) "...the pattern of neuronal activation by somatic isolation was distinct from that associated with reunion..." Also, the assumption that amylin acts as a retrograde messenger is highly speculative and appears to rest solely on a histochemical observation. Furthermore, the authors (scholarly) describe that it remains not known with much certainty that the calcitonin receptors is activated by amylin in the brain, which makes the proposed chain of events weaker. To strengthen this argument, future experiments should be directed at showing that this is indeed the case, by e.g. activation of CalcR neurons by local amylin or by optogenetic activation of amylin neurons. – Regrettably, the Discussion fails to connect the findings where cause-effect relationships are convincingly demonstrated to the reader. This section also strays from the actual data quite a bit by e.g. at length discussing (p.18) the role of oxytocin, a peptide whose relationship to the actual data is minimal. Finally, the argument that the same system sustains maternal care and affiliative behaviour towards conspecifics relies on data in an unpublished manuscript (not made available as a preprint it seems). Thus, this important aspect of the work is difficult to evaluate.

This is a technically largely impressive study with significant individual strengths. But the dots are not connected to a complete picture that convincingly clarifies the role of the amylin MPOA system in social interactions to the reader.

MINOR CONCERNS

I suggest moving the initial pg in Results (lines 96-104) to the Introduction, where it makes more sense. The relationship between amylin and CalcR also needs to be made clearer to the reader.

Line 33: "recover" should read "recovery"

Lines 34-35: novel and conserved? What is the basis for this claim?

Line 126-127: "...nor different palatable foods every day could not induce Amylin mRNA expression in isolated females..." Awkward grammar; a double negative?

Lines 146-9: "It turned out that, in-cage isolation by wire-mesh walls (designated as "somatic isolation", Fig. 1p, to differentiate from complete isolation = single housing in a cage. Both types of isolation are correctively called as social isolation.) resulted in a loss of Amylin mRNA expression (Fig. 1q)." The use of full sentences within parenthesis of another sentence is confusing and, to this reviewer's knowledge, incorrect according to common rules of writing. Please rephrase. I am also not sure I understand how the authors use the term "correctively"?

Line 308 "immediately" should read "immediate"

I failed to find information on how the primary antibodies have been validated. Can the authors please explain where this is included or, if it is not, please add this important information?

Lines 1212, 1231, 1254, 1278, 1302, 1326: "statics" should read "statistics"

Fig. 1: The schematics are not sufficient to understand how the different paradigms were organized (e.g. wire mesh cage within cage etc). A photograph of better resolution would help clarify these points.

Fig 1p: "Samatic" should read "Somatic"?

Reviewer #3:

Remarks to the Author:

In their study, the authors found that a particular population of amylin-expressing neurons in the medial preoptic area (MPOA) are involved in affiliative social behavior in mice. They first show that not only complete but even partial social isolation of mice (by means of restricting an individual in

a wire cage within the group housing cage) leads to loss of amylin expression in MPOA. They then show that complete and partial social isolation produce behavioral stress responses as well as increased c-Fos expression in the paraventricular nucleus of hypothalamus (PVN). Subsequently they show that reunion with peer mice after previous social isolation resulted in particular affiliative social behaviors, while c-Fos expression in PVN was not increased. Next, the authors quantified c-Fos expression in various "social" brain structures (they chose amygdala and bed nuclei of stria terminalis (BST) in addition to MPOA) after partial social isolation and found increased activity in various anxiety-related subnuclei of these structures such as the central amygdala, while no increased activity was found in the preoptic area (including MPOA). They did the same after social reunion and found strikingly different patterns of c-Fos labelling, specifically in the preoptic area, as well as some other areas such as medial amygdala. Next, the authors showed that a probably receptor for amylin, calcitonin receptor (Calcr), is expressed in MPOA as well as some other social forebrain regions such as BST and medial amygdala, revealing a general pattern that was notably similar to the c-Fos labelling after social reunion. When they performed anti-Calcr stainings of the c-Fos material of social reunion mice, they found colabelled neurons especially in MPOA, and further found that these were vGluT2-expressing. In contrast, social isolation mice did not reveal Calcr+/-Fos+ colabelled neurons. Next, the authors tested the effects of excitotoxic NMDA-lesions of MPOA on social isolation and reunion behaviors and found that biting of the separation wall in partial social isolation experiments as well as social contacts and sleep huddle in reunion experiments were reduced. To test whether this was due to Calcr-signalling in MPOA, they performed RNA interference-knockdown of Calcr in MPOA using shRNA against Calcr introduced via an AAV. In partial social isolation experiments, this knockdown led to attenuated "protest" behaviors against the partial isolation, as well as reduced contact-seeking behaviors after social reunion, while long-term contact behaviors and sleep huddle were not inhibited, indicating a role of Calcr MPOA neurons in acute-phase social isolation/reunion behaviors. The authors then investigated the relationship between Calcr neurons and amylin neurons. Using a novel Amylin-Cre mouse which the authors constructed, they found contrary to intuition no Amylin+-synapses close to Calcr+somata, but rather Calcr+-synapses close to Amylin+-somata, and concluded that this might be a case of retrograde signalling. The authors next checked sexual dimorphism of the amylin / Calcr system and found that males had fewer of those neurons and were less afflicted by partial isolation as well as showed more aggression upon social reunion. Ultimately, the authors performed activating-DREADD experiments in central amygdala CaMK2a neurons, finding that social motivation behavior during partial isolation was increased.

In general, this is a very interesting study full of interesting findings and beautiful data. The study offers a broad basis of methods which were elegantly used by the authors to investigate their specific scientific questions. However, as my summary of the whole manuscript above already shows, parts of the study appear to be juggled together, which leads to a loss of the focus of the study. For example, the last two parts of the study (and the pertaining figures 8 and 9) seem out-of-place. I think their main results could be integrated into the manuscript at previous positions, and their figures strongly condensed (with parts moved to the supplements). Right now, the manuscript at times appears too bloated. This is also indicated by some of my following specific comments and with respect to other parts of the manuscript. I would like to propose to the authors to revise the manuscript, put more focus on the most relevant aspects and try to integrate the not-so-relevant aspects more smoothly (or, if they do not fit, remove them from this manuscript). I am sure that after a thorough revision, this will be a very nice paper suitable for publication in Nature Communications.

Specific comments:

- At several points, the authors distinguished between social reunion (as affiliative behavior) and "defensive huddle" (as social stress coping behavior). However, to induce defensive huddle they just exposed some mice to very bright light, which means that any effects observed probably result from general stress rather than the actual social coping behavior. All parts relating to "defensive huddle" behavior should be removed from the manuscript. This would in my opinion increase the quality of the study.
- I see a small weakness in the lack of conceptual interpretation of the diverse behaviors which the authors have identified and selectively scored. While I think that the particular distinguished behaviors were well selected, I would recommend some simplification in the text: For example, I would summarize the diverse behaviors under overarching terms such as "body contact behaviors"

and "social motivation behaviors" (the latter e.g. referring to all kinds of bar biting and rearing etc. in partial isolation experiments, which just indicate that the isolated mouse really wants to get through the bars back to its peers). This would make the text easier to understand in my opinion.

- The Methods section in particular requires English proofreading. The whole manuscript would in fact greatly benefit from native English proofreading.

- Line 33: "recover" should probably read "recovery"
- Line 35: Why conserved? Could be, but this is not really discussed in the manuscript.
- Line 47: hypothalamus-adrenal-pituitary should read hypothalamic-pituitary-adrenal
- Line 102: delete "the"
- Line 106: first-time mention of brain structures, provide full names
- Line 120: The "restraint stress" paradigm is not described in the Methods section. Please add this and elaborate.
- Line 125 ff: change to "however, neither environmental (...) were sufficient to induce (...)"
- Line 176 ff: The term "real partition" refers to an insertable wall with bars in contrast to a wall with open doors ("sham partition"), but these terms are confusing, especially in view of the previously applied wire-cage-within-group-housing-cage paradigm. First, talking about "partition walls" instead of just "partitions" would slightly improve understandability. Second, I think that the "sham partition" concept is not relevant, as mice can freely move through the doors as if there was no barrier.
- Line 196 ff: The term "Somatic isolation" does not really make sense. Instead, I would suggest using a term such as "partial isolation". This would also have to be corrected in figures, such as Fig. 1.
- Line 200-202: Please improve English grammar.
- Lines 196-253 (describing results in Fig. 2) and 1216-1231 (Fig. 2 incl. legend): In great detail, behaviors resulting from complete or partial ("somatic") isolation vs. group-housing are described. I have several problems with this part: 1) it appears to lose focus of the main findings and purpose of the study, so it should be greatly condensed and partly moved to the supplementary material. 2) Main focus of Fig. 2 and the accompanying text should be the finding that both complete and partial isolation result in stress behaviors. 3) I do not understand the relevance of the "two-stage response" to social isolation (line 1216) in the scope of this study.
- Lines 255-286: Again, too much focus on behavioral details resulting from the reunion of previously socially isolated mice. These observations, while in themselves interesting, do not seem of relevance to the purpose of the study.
- Line 285: The authors claim that defensive huddle involves a PVN-mediated stress reaction. However, this is a non-sequitur: It is more probably that the bright light which they used in this paradigm by itself caused the stress-induced increase in PVN activity. In any case, the whole point seems irrelevant to the purpose of the study.
- Lines 319-327: Same problem as previous comment, the bright light probably induced stress which resulted in the observed c-Fos patterns, and not the "defensive huddle" behavior as claimed.
- Lines 430-436: To test amylin function onto Calcr neurons in social behaviors, the authors injected the Calcr antagonist AC187 into the cerebral ventricles and found some effects on social contact behaviors. However, I wonder given the many caveats this method carries (such as: does the molecule actually reach the Calcr neurons by diffusion? Which Calcr neurons in which regions are inhibited? Etc.), why the authors did not perform direct infusion of AC187 into MPOA via implanted guide cannulas as in (e.g.) Hasan, Mazahir T., et al. "A fear memory engram and its plasticity in the hypothalamic oxytocin system." *Neuron* 103.1 (2019): 133-146.
- Lines 453-465: Why did the authors focus exclusively on CeM CamK2a neurons? I find their short justification given in the text too little. Especially, since this part of the study appears entirely disconnected from the actual topics of Amylin and Calcr.
- Figure 1a: Not clear to the reader why NPI was used as second antigen in this staining
- Figure 1 legend:
 - o In lines 1203-1205 there is a problem, these lines need to be redacted entirely.
 - o In general, there are many small grammar problems in this legend, please proofread.

Referees' comments and our point-by-point responses

*Please note that, in the revised main text, we used color-coding: **Red** colored changes were for Reviewer #1's comments, **blue** for Reviewer #2's, and **green** for Reviewer #3's comments. Other changes were **orange brown**.

REVIEWER COMMENTS

Reviewer #1 (Remarks to the Author):

The present study by Fukumitsu et al. is a unique and important investigation on how the calcitonin receptor signaling in the MPOA mediates affiliative social behavior in adult female mice. They recognize the relevance of performing most of these experiments in female mice, escaping the restrictions of studying affiliative social behavior in males (due to intermale aggression). They use robust behavioral paradigms, and through clear quantifications, they suggest the significance of Amylin mRNA expression in social behavior. I have no doubt that this manuscript will be a significant addition to the literature.

It is important to note, however, that this manuscript requires attention from the authors and a major revision. Despite the wealth of data, I often found the writing and the flow of concepts in the text and figures confusing. Below I provide an outline of the points I would like to see clarified, with the aim that future readers can appreciate better the content of this work.

Response: First, we truly appreciate your precious time and effort, positive assessment of this manuscript, and pointing out key issues for improvement. Texts were colored in **red** for revisions pertaining to your comments. We also tried to improve the text and figures so that they are not confusing, as detailed below.

Major points

1. The authors make a case that the use of females is a better option for the study of affiliative social behaviors. They make a case for this in the Introduction (lines 64-85), but I would like to see this rationale reinforced. The question "Why study this on females?", needs to be explicitly clarified to the reader.

Response: We agree that the paragraph in the Introduction describing the benefit of studying females for social behaviors was not compelling enough. We re-organized this paragraph

into two paragraphs, and added following sentence "Taken together, as previously suggested, it is not only beneficial, but also indispensable, to use female subjects to elucidate the neural mechanism of sociability and social affiliation^{25,35}." (page 4).

The authors also cite literature on maternal care – which I understand that they do because of the other study they have on review with similar findings (citation 47). In the introduction, however, I found such text and references distracting. The point of the present manuscript is to present an investigation on the neuro-molecular mechanisms that drive affiliative behaviors among adult mice. I would suggest only slight deviations for this line, by citing literature in maternal care etc.

Response: We agree with this comment and have omitted one paragraph about maternal care from the introduction.

Meanwhile, the original citation 47 (Calc-amylin system on maternal care) has now been published in Cell Reports, thus the relevant descriptions were modified in this manuscript.

2. Have the authors considered the effects of the estrus cycle in their behavior readout? This is important. What are the behavioral responses of ovariectomized mice in the social isolation test (presented in Figure 2)? What are the behavioral responses of ovariectomized mice supplemented with chronic estrogen-releasing mini-pumps in the same test?

Response: Thank you for pointing this important issue. We have performed the behavioral test using ovariectomized females and found that the contact-seeking, biting behavior after somatic isolation was reduced by ovariectomy. We also found that the supplement of E2 could not completely recover this defect caused by ovariectomy. Of note, we have previously reported that the amylin mRNA expression is strongly dependent on estrogen and prolactin, and that E2 supplement could only partially rescue amylin expression (Fig. 5 of (Yoshihara et al., 2021)). Thus, the decrease of contact-seeking following ovariectomy can be attributed to the decrease in amylin expression in the cMPOA. We have described these findings in the revised text (page 4) and in the supplementary figure S8.

3. How do the authors explain that the Amylin peptide – known for its role in energy metabolism and anorexigenic effect, has a role also in social behavior?

What is their understanding of the interplay between energy metabolism, and social behavior, through what is seemingly common mechanistic substrates?

Are they studying the effect of energy metabolism-related changes on behavior, or is

Amylin a common substrate for both food intake AND social behavior? The reader would benefit greatly from understanding this, and the inclusion of such text would be relevant in the Introduction and/or Discussion.

Response: As the reviewer keenly points out here, there is a strong functional connection between the regulation of social behavior and energy metabolism. We have discussed this issue in the Discussion as follows:

"It remains to be clarified why amylin and its receptor CalcR-RAMP complex, the well-known regulatory system for appetite control {Young, 2005 #2520} {Lutz, 2006 #2521}, should also mediate affiliative social contacts and parental care. Of course, it is possible that amylin is used purely as a signaling component, and independently regulates food intake in the area postrema and social behavior in the cMPOA. Another possibility is that the level of amylin indeed conveys metabolic information along with social information to collectively regulate social behaviors. It has been known that the social organization of house mouse *Mus musculus* is highly variable, ranging from being solitary in non-commensal populations (i.e., inhabitant in fields, sand dunes), to forming high-density multimale/multifemale colonies in commensal populations (i.e., inhabitant in buildings or human settlements) with superabundant food supply {Poole, 1985 #2593}. Mice in a high-density commensal population are known to be less aggressive and more amicable {Frynta, 2005 #2480}. Notably, amylin produced in the hypothalamus are reported to be scarce in males but are dramatically increased by high fat diet {Li, 2015 #2649}, thus it is reasonable to speculate that increased amylin by satiety could induce sociability in mice. On the contrary, food scarcity is one of the major limiting factor of group size, and individual animals increase solitary foraging to avoid intragroup food competition. In such cases, the limited amylin binding to CalcR in the cMPOA may downregulate contact-seeking behaviors to better adapt the environment. Future direct experimental demonstration is needed to prove this appealing possibility." (page 16)

4. I observed a general tendency by the authors to subjective/anthropomorphic interpretation of the data across the manuscript. While I cannot provide an exhaustive list of this, here is one example.

line261: "female mice gradually calmed down"

I urge the authors to report quantification of behavioral parameters, and not include writing indicative of a subjective interpretation of the data - especially in the Results section.

Response: We agree with this comment and have revised the manuscript accordingly as noted below and in the replies to the minor comments for lines 48 and 153-155 from the reviewer

#1;

"the "protest" phase" -> " the **initial active** phase" (line 225)

"the "despair" phase" -> "the **following inactive** phase." (line 226)

"female mice gradually calmed down" -> "female mice gradually **became inactive**" (line 246)

"formed a cluster and slept together, called sleep huddle," -> " **formed a cluster termed a sleep huddle,**" (line 246)

5. A methodological point: regarding the use of a mesh for partitioning cage compartments, I suggest to the authors the use of a plastic (PVC or equivalent) mesh instead. Mice will naturally exhibit biting behavior towards a metal mesh with round wires, especially upon the initial introduction of such a mesh to their cage – when the mesh is a novel object. Have the authors habituated their mice to this material prior to its use in the experiments of Figure 2? Could this affect the behavioral readouts presented in Figure 2?

Response: All the subject mice were habituated with the same-metal, sham partition with biteable windows at least for 6 days (Fig. 2a), so that the metal partition was not new for the mice. Then, all the mice were exposed to introduction of the real partition, resulting in a slight increase in biting behavior in all group of mice (Fig. 2e, peaking at 30 min timepoint). However, there is a significant difference between the somatic isolation group and the other groups. This experimental setting should exclude the possibility that the somatic isolation-induced biting behavior is caused by the general tendency of metal biting in mice.

6. This study uses a number of uncommon terms that need to be explicitly defined. I strongly recommend to the authors that when they use such terms, they use writing similar to the following rephrased example (relevant to lines 260-263):

“We often observed that mice, following reunion, formed a cluster and remained inactive or asleep, a behavior we defined as sleep huddle. Sleep huddle has an obvious role in thermoregulation...”

Such clear definitions should be provided in the main text for:

a) defensive huddle, b) protest behavior, c) despair behavior, d) full social contact, e) somatic isolation

Response: The terms "protest" and "despair" were used extensively in classic literature on maternally-separated primate infants (Bowlby, 1969; Lewis, 1976). Along with our revision condensing descriptions about maternal separation, these terms were now completely removed from the text. For explanation of current data, we replaced "protest" to " **contact-seeking and partition biting** ", and "despair" to " **following inactive phase of hunched posture**".

- a) " here we also characterized a distinct situation that involves significant physical contact among cage mates, called a defensive huddle (Bowen et al., 2013). A defensive huddle is defined as aggregation of individual animals as a strategy of self-defense under a potential threat. " (page 9, lines 250)
- b) removed
- c) removed
- d) full social interaction -> free social interaction (Abstract); Full social contact -> free social interaction (page 7)
- e) " to specify different kinds of isolation, we termed single housing in one cage as "complete isolation" (Fig. 2b), and use "somatic isolation" to describe a subject mouse housed in the same cage with other mice but separated by a coarse mesh or windowed partition as in Fig. 2a and Supplementary Fig. S1f. "(Social) isolation" includes both complete and somatic isolations." (Page 7)

7. Throughout the Figures, the authors place the Y axis label in all graphs, in the place were one commonly places the title of the graph. Place the Y axis label, next to the Y axis in all figures.

Response: We have placed the Y axis label next to the Y axis in all figures.

8. If I understood correctly, the authors use C57 mice for the data presented in Figure 1, and BALB/c for the data presented in all other figures. I suggest that the authors move Figure 1 to the Supplementary material, and repeat the experiments presented in Figure 1c-e, and Figure 1p, q in BALB/c mice. All main figures, should be presenting data from the same mouse strain.

Response: As suggested, we repeated the most critical experiments of Fig. 1c-e and Fig. 1p, q in BALB/c mice (newly added Supplementary Fig. 2e-f, and g-h, respectively), confirming essentially the same regulation of amylin expression by social contacts in both mouse substrains.

While placing all the other panels in the Fig. 1 into the supplementary material as suggested is possible, we still think that it is meaningful to show the other data, such as stress-insensitive Amylin expression in Fig. 1i-j, activation by co-housing and deactivation by social isolation of amylin+ neurons in Fig. 1k-n, and the inability of olfactory or other distal sensory cues to maintain Amylin expression in Fig. 1o, p-q, in the main text. This line of experiments clearly show how we have worked and shaped the logic for the following experiments. If all of them were in the supplementary figure, we are afraid the relevant parts of the text become difficult to read. Redoing all these experiments in BALB/c is theoretically possible, but it will not only take considerable time and money, but also a huge amount of animal life. Therefore, in the revised manuscript we still include the entire C57BL/6 data in Fig. 1 and include the key BALB/c

replication in Figure S2; we look forward to hearing the reviewer's and the Editor's opinion on this issue.

9. Perhaps I missed the point of this, but to my understanding Figure 9 does not belong to this manuscript. I found it irrelevant and superfluous, and therefore suggest it's exclusion from the revised version.

Response: We agreed with this comment and removed the original Fig. 9 from this manuscript.

Minor comments

Relevant to Major Point #6. In the Introduction, the authors cite nomenclature I am unfamiliar with, such as "protest behavior" and "despair phase". The sentences containing these concepts have no citations. Are the authors defining new concepts at this point in the manuscript, or are they using these concepts from previous literature. Please clarify.

Response: The "protest" and "despair" has been used in previous literature in the context of maternally-separated primates (for example, Bowlby "Attachment", 1969). We have added the citation, but removed the details about these concepts. (page 8).

Figure 1p: change samatic isolation to somatic isolation

Response: Corrected accordingly.

line 46: change "sociable animals" to "social animals". Consider performing this change throughout the manuscript.

Response: Actually, using "social" as an adjective for animal species has been cautioned by the fact that sociality consists of at least three independent facets (Poole, 1985) {Kappeler, 2002 #2603} (see also (Caldwell and Albers, 2021)). Thus we introduced a more specific term "sociable" instead, to indicate ""non-solitary" species {Poole, 1985 #2593} possess the tendency the tendency for adult animals to stay together for a certain period of time in an affiliative manner, but not merely for reproductive purposes or for agonistic/competitive purposes". The explanation of these terms was originally in the Supplementary Discussion, but according to this comment, we added this definition in the beginning of the Introduction (page 3).

line 48: change “mood” to “behaviors”. Identifying the mood of mice is difficult, but tracking their behavior is not.

Response: Thank you for pointing this out, we have revised the text accordingly.

(Original) When separated from peer conspecifics, these sociable animals may exhibit stress responses such as an activation of the hypothalamus-adrenal-pituitary (HPA) axis and depressive mood

(Revised) When separated from peer conspecifics these sociable animals may exhibit a two-stage response, with initially enhanced and later depressed physical activity and stress responses, such as activation of the hypothalamic-pituitary-adrenal (HPA) axis

line 52: change “(for example, 11)” to “(for example, Zelikowsky et al.11)”

line 55: correct the notation of citations (“ 12-18 19 ”)

Response: Corrected accordingly.

line 62-63: check whether this sentence “However, it remains poorly investigated if isolation induces such a series of acute reactions in adult mice.” Should be rephrased as “However, it remains poorly investigated if isolation induces such a series of acute reactions in mouse pups.”

Response: We originally meant that acute reactions were investigated and found in pups and infants, but not much in adults. However, this sentence was removed along with the explanation about maternal separation literature (comment #1).

line 73: change “sociality” to “sociability”

Response: Thank you, we changed this word accordingly.

line 80, 81: “With such ethological considerations, it is not surprising that male mice may not exhibit depression by social isolation but rather show stress responses to group housing 44 25 45.”

Reviewer: I do not agree with this sentence, single housing is stressful for male mice, as the author note on line 52 and by citing Zelikowsky et al.

This sentence and concept require revision.

Response: We have toned down the sentence as follows; "Therefore, it is not surprising that male mice may exhibit **fewer isolation-induced depressive behaviors than females** and rather, can show stress responses to group housing^{44 25 45}" (page 3).

Line 121: "A two-way ANOVA revealed...". Consider rephrasing this sentence. A statistical test permorms a comparison between two data sets. The experiment revealed your finding.

Response: Agreed. We have modified this sentence "**This experiment revealed** that ---".

Line 127: The authors discuss in the text that environmental enrichment did not induce higher mRNA expression in isolated females, but they don't show the data. I recommend that this data set is included in the revised version as a supplementary figure.

Response: We added Supplementary Figure 1a-c to show this data.

Line 136: Ambiguous phrasing. What is a "full social contact"? Following this line, the authors define what they mean by somatic isolation, and the reader can infer what a full social contact means. However if they want to use this phrasing, they have to define what a full social contact is prior to using the term in the text.

Response: We changed "full social contact" to "**free social interaction**", meaning the contact is not limited by a mesh partition.

Line 144: change "could not able to" to "did not"

Response: Corrected accordingly.

Lines 146-149: rewrite sentence, hard to read/understand

Response: We apologize for this awkward sentence and revised as follows; "**Identical to the effect of single-housing, in-cage isolation resulted in a loss of *Amylin* expression after 6 days (Fig. 1q, Supplementary Fig. S2h).**"

We moved the inserted explanation of somatic isolation and complete isolation to the text referring to Fig. 2 as described above (page 7, lines 166).

Lines 153-155: "The isolated female, however, seemingly perceived social isolation; soon

after containment in the compartment, the female vigorously bit, dug beneath, and climbed up the wall, apparently trying to reunite with conspecifics.”

Here the authors suggest the intent of the mouse. That is, that it bit the mesh, dug beneath, and climbed up the wall to reunite with conspecifics. Scientific experiments and writing should not allow for such subjective interpretations over why a mouse behaves a certain way. The field of systems neuroscience uses behavioral tests, and studies report behavioral quantifications when using these tests. Writing of this type does not belong to scientific articles, not in the Results, and neither in the Discussion.

Response: We agreed with this comment and removed the descriptions; "female mice recognize and seek for free interaction with conspecifics", and "Female mice apparently trying to reunite with conspecifics". The remaining part was moved to the Supplementary Fig. 1 legend in the Supplementary material.

Lines 227-229: “separated BALB/c adult female mice exhibit a two-stage separation distress response consisting of the first active "protest" phase of about 45 min, followed by the passive "depressive / despair" phase”. These phases were presented in the introduction as behavioral phases of pups or infants. Here the authors use the same terminology in adult mouse behaviors.

Response: We removed these terms from the text (comment #6). Just for the reviewer’s information, these terms were originally coined for infants and later extended to peer-separated adult animals (McKinney, 1985).

Authors often write in the legends (n = # per group). Please specify units. Do you refer to mice, sections, etc.?

Response: Throughout this manuscript, the sample numbers meant the number of mice. We clarified this in the legends.

Figure 1a. Present the images at larger size in the figure.

Response: We have modified the figure accordingly.

Figure 1b. Are there any significant differences between brain areas? Illustrate this in the graph. In the legend of Figure 1b, it’s written (n=5). 5 mice, 5 sections per mice? Please

clarify.

Response: We added the statistical analysis for Fig. 1b. All the sample size are the number of mice (biological replicates) in this study.

Figure 1d. Increase the size of the images. Increase the contrast of the images, so as the immunoreactive cells become more visible.

Figure 1g. do the same as suggested for Figure 1d.

Figure 2m: Increase the size of the images. Increase the contrast of the images, so as the immunoreactive cells become more visible.

Response: We modified the images accordingly. Thank you very much again for your precious advice.

Reviewer #2 (Remarks to the Author):

In this study, Fukumitsu et al. have characterized the behavioural sequence of isolation and reintroduction of company in female mice, as well as its correlates in brain activation, with a focus on the amylin-calcitonin receptor system. Amylin neurons in the MPOA are shown to be sensitive to social conditions, as amylin expression is strongly depleted by isolation, and recovered by reunion with peers. Attenuation of signalling in this pathway by gene knockdown or pharmacological antagonism diminishes the amount of interaction during reunion.

This is an ambitious study on an intriguing topic; there is much work currently being published on the brain substrates for social behaviour, but the hypothalamic amylin system has received limited prior attention. Substantial effort is here made to work out the protocol for social isolation, which I would expect will be adopted by other investigators in the future. The authors have been careful to include appropriate controls, to determine which aspects of contact with peers cause changes in neuronal activation. Taken separately, each (or most) of the experiments are technically solid, and the level of detail and insightful description of methods is likely to be of great value to many others in the field as they design their experiments. (There are some drawbacks, however: e.g. the use of NMDA lesions is inferior in precision for targeting a particular population compared with genetically based approaches.) This is a great strength of the work.

Response: We really appreciate your precious time for evaluation and pointing important issues required for improvement of this manuscript. The changes made for your comments were

color coded by **Blue**.

At your suggestion we have reduced or removed the mentions about experiments using inferior techniques, such as the NMDA lesion or i.c.v. injection, and added more specific experiments to strengthen the manuscript in the new Fig. 8.

The weakness of this study is that the data from the individual experiments are not coherently connected and the proposed amylin system is based largely on correlations. As some examples, it is not clear what the significance is of the demonstration that (lines 328-9) "...the pattern of neuronal activation by somatic isolation was distinct from that associated with reunion..."

Response: We agree with this comment, and have rephrased the sentence into more careful description.

(Original) the pattern of neuronal activation by somatic isolation was distinct from that associated with reunion

(Revised) **no overlap was found between the brain subregions significantly activated by reunion and those activated by somatic isolation**

Also, the assumption that amylin acts as a retrograde messenger is highly speculative and appears to rest solely on a histochemical observation.

Response: We have removed the speculation of possible retrograde amylin transmission and shortened the relevant part as follows;

(Original) This configuration of these two types of MPOA neurons suggests that Calcr+ neurons are presynaptic and send inputs onto amylin+ neurons, and postsynaptic amylin signals back to Calcr+ neurons. This type of transmission from post- to pre-synapse is called as a "retrograde signaling" and has been shown in opioid, anandamide or nitric oxide. As in C57BL/6J virgin females {Yoshihara, 2021 #2554}, the majority of Calcr+ neurons in the cMPOA were glutamatergic, and the majority of Calcr+ neurons in the ACN were GABAergic in BALB/c virgin females (Fig. S4); thus these classical neurotransmitters are most likely secreted from Calcr+ neurons to activate/suppress excitability as well as amylin secretion of amylin+ neurons.

(Revised) This configuration suggests that Calcr+ neurons might be presynaptic and amylin **might signal back to Calcr+ neurons. This type of neuromodulation has been reported** in opioid, anandamide or nitric oxide.

Furthermore, the authors (scholarly) describe that it remains not known with much certainty that the calcitonin receptors is activated by amylin in the brain, which makes the proposed chain of events weaker. To strengthen this argument, future experiments should be directed at showing that this is indeed the case, by e.g. activation of CalcR neurons by local amylin or by optogenetic activation of amylin neurons.

Response: In response to this comment we performed the indicated experiment of local application of amylin (Fig. 7e-h) and demonstrated the significant activation of CalcR neurons by amylin. Moreover, the addition of new experiments of chemogenetic activation of Amylin+ neurons and *Amylin* gene targeted mice consistently showed that the function of amylin on CalcR+ neurons is to induce contact-seeking biting behavior at somatic isolation. Therefore, the revised manuscript provides a much more complete story about the causal role of the amylin-CalcR circuit to regain social contact in female mice.

This section also strays from the actual data quite a bit by e.g. at length discussing (p.18) the role of oxytocin, a peptide whose relationship to the actual data is minimal.

Response: We agree that the original oxytocin-related paragraph was unnecessarily long, and shortened the oxytocin-related discussion as follows.

(Original) It should be noted, however, that the effects are only partial, indicating that the other factor(s) outside of the amylin-CalcR signaling take a significant role, in particular, oxytocin. Oxytocin has long been implicated in various kinds of social behaviors, in both affiliative and agonistic/competitive components {Insel, 2001 #156;Goodson, 2013 #2602;Bosch, 2017 #2624;de Jong, 2018 #2617;Jurek, 2018 #2623; Bredewold, 2018 #2631}, although the role of oxytocin in maintaining parental nurturing behaviors may be rather regulatory {Yoshihara, 2017 #2086}. Even limited to non-reproductive social contexts in non-human mammals, there are several studies that have demonstrated that oxytocin neurons are activated by social stimuli, and facilitate social preference and/or contacts {Okabe, 2015 #2619;Oettl, 2016 #1583;Hung, 2017 #2620;Resendez, 2020 #2627;Tang, 2020 #2579}. Recently Tang and colleagues have demonstrated that PVN oxytocin neurons are activated specifically during body-trunk contacts, in particular on the back, but not during sniffing, chasing, vocalization (of self or the counterpart), or head-to-head contacts. They have also shown that parvocellular oxytocin neurons facilitate social contacts, especially "crawling on the top" behavior in female rats. Our study confirmed their important findings that tactile sensation obtained by free social interaction, but not by "chambered social interaction" (similar to our "somatic isolation"), is the key component of social interaction among females. It should be noted that when the activation of oxytocin neurons is detected by c-Fos expression, quite a small (1~3 %) population of total oxytocin neurons become c-fos positive by gentle stroking or

non-noxious mechanical stimuli (air-puffs) on the back {Okabe, 2015 #2619;Tang, 2020 #2579}. Consistently, we could barely detect c-Fos expression in the oxytocin neurons in the ACN after reunion (Fig. S11). It is found that amylin expression is highly dependent on prolactin and estrogen, but not on oxytocin {Yoshihara, 2021 #2554}. Considering that the effects of oxytocin knockdown or neuronal silencing on social contacts are also partial, it is probable that both amylin-Calcr and oxytocin contribute to social contact behaviors independently.

(Revised) It should be acknowledged that the other factor(s) outside of the amylin-Calcr signaling should also take a significant role in facilitating social contacts, as the blockade of amylin-Calcr signaling exerts only partial effects on social contact behaviors. Oxytocin neurons has been demonstrated to be activated by social stimuli, and facilitate social preference and/or contacts in non-reproductive contexts {Lukas, 2011 #2928; Okabe, 2015 #2619; Oettl, 2016 #1583; Hung, 2017 #2620;Resendez, 2020 #2627;Tang, 2020 #2579}. Tang et al. has shown that somatosensation at the body-trunk by free social interaction as well as by air-puffs, but not "chambered social interaction" (similar to our somatic isolation) activates oxytocin neurons in the PVH, and chemogenetic activation/inhibition of oxytocin neurons facilitate/inhibit social interactions among female rats, respectively. Because amylin+ and Calcr+ neurons are distributed together with oxytocin neurons in the ACN, it is possible that amylin-Calcr neurons may act in concert with oxytocin neurons to sense and regulate affiliative social contacts.

– Regrettably, the Discussion fails to connect the findings where cause-effect relationships are convincingly demonstrated to the reader.

Response: The Discussion has been thoroughly revised to describe the newly added causal effects of amylin-Calcr in contact seeking behaviors, as detailed in the last major concern of this reviewer (please see below).

Finally, the argument that the same system sustains maternal care and affiliative behaviour towards conspecifics relies on data in an unpublished manuscript (not made available as a preprint it seems). Thus, this important aspect of the work is difficult to evaluate.

Response: The mentioned manuscript by Yoshihara et al. has been published, and the relevant reference was updated.

This is a technically largely impressive study with significant individual strengths. But the dots are not connected to a complete picture that convincingly clarifies the role of the amylin MPOA system in social interactions to the reader.

Response: We agree that in the original version there were missing links connecting each figure together into a convincing story, like whether amylin is indeed the ligand of Calcr. To fill these gaps we have performed three new experiments to directly demonstrate that the amylin can influence Calcr+ neurons via Calcr molecules and regulate social contact behavior.

Briefly, to bidirectionally show the causal role of the amylin system in social contact regulation, we have performed an additional *downregulation* experiment using a genetic mutant of amylin gene (Amylin KO) as shown in new Fig. 8f-k, and another *upregulation* experiment using chemogenetic activation of Amylin+ neurons as shown in Fig. 8a-e. Together with Calcr knockdown experiment in Fig. 6, these experiments consistently demonstrated Amylin-Calcr function in inducing contact seeking behaviors. And for the experimental evidence whether amylin can activate Calcr, we performed amylin microinfusion into the cMPOA and confirmed transcriptional activation of Calcr+ neurons in Fig. 7e-h. We think that these results are clear-cut and fill the missing logical link pointed out by the Editor and Reviewer #2.

MINOR CONCERNS

I suggest moving the initial pg in Results (lines 96-104) to the Introduction, where it makes more sense. The relationship between amylin and Calcr also needs to be made clearer to the reader.

Response: We totally agree with this comment, thank you. We moved the first paragraph of the original Results into the last part of the Introduction, and added more background information about the relationship between amylin and Calcr in the last paragraph of the Introduction.

Line 33: “recover” should read “recovery”

Response: We corrected this word accordingly.

Lines 34-35: novel and conserved? What is the basis for this claim?

Response: Sorry for this unclear phrasing, we have revised the sentence as follows;

(Original) These data present a novel and conserved neuromolecular basis of mammalian affiliative behaviors

(Revised) **As the same amylin-Calcr signaling is essential for maternal care, the present data also provide neuromolecular evidence for the long-postulated common origin of social affiliation and maternal care in mammals.**

Line 126-127: "...nor different palatable foods every day could not induce Amylin mRNA expression in isolated females..." Awkward grammar; a double negative?

Response: We corrected this sentence as follows;

"However, **neither** environmental enrichment by providing attractive non-social stimuli (e.g. a running wheel, novel objects as a toy), **nor different palatable** foods every day **were sufficient to induce** *Amylin* mRNA expression in **single-housed** females (**Fig. S1a-c**). "

Lines 146-9: "It turned out that, in-cage isolation by wire-mesh walls (designated as "somatic isolation", Fig. 1p, to differentiate from complete isolation = single housing in a cage. Both types of isolation are correctively called as social isolation.) resulted in a loss of Amylin mRNA expression (Fig. 1q)." The use of full sentences within parenthesis of another sentence is confusing and, to this reviewer's knowledge, incorrect according to common rules of writing. Please rephrase.

Response: We corrected this awkward part as, " **It turned out that, in-cage isolation by wire-mesh walls resulted in a loss of *Amylin* mRNA expression, just as single-housing (Fig. 1q).**" We moved the inserted explanation of somatic isolation and complete isolation to the text referring to Fig. 2 (page 7, lines 166).

We also modified other sentences according to this comment, for example, the second sentence in the Introduction.

I am also not sure I understand how the authors use the term "correctively"?

Response: Sorry for this mistake, it was changed as "collectively".

Line 308 "immediately" should read "immediate"

Response: We corrected this word accordingly.

I failed to find information on how the primary antibodies have been validated. Can the authors please explain where this is included or, if it is not, please add this important information?

Response: Sorry for omission, but information about primary antibody validation shall be included in "Reporting Summary", a separate form for this journal. Briefly, all the primary

antibodies used in this study are popularly-used commercial antibodies and are validated by each vendor.

Lines 1212, 1231, 1254, 1278, 1302, 1326: “statics” should read “statistics”

Response: The wrong word "statics" has changed into "statistics" throughout the manuscript.

Fig. 1: The schematics are not sufficient to understand how the different paradigms were organized (e.g. wire mesh cage within cage etc). A photograph of better resolution would help clarify these points.

Response: We have added photographs in Supplementary Fig. S1f.

Fig 1p: “Samatic” should read “Somatic”?

Response: We have corrected the word.

Again, please accept our gratitude for your precious advice and comments.

Reviewer #3 (Remarks to the Author):

In their study, the authors found that a particular population of amylin-expressing neurons in the medial preoptic area (MPOA) are involved in affiliative social behavior in mice. They first show that not only complete but even partial social isolation of mice (by means of restricting an individual in a wire cage within the group housing cage) leads to loss of amylin expression in MPOA. They then show that complete and partial social isolation produce behavioral stress responses as well as increased c-Fos expression in the paraventricular nucleus of hypothalamus (PVN). Subsequently they show that reunion with peer mice after previous social isolation resulted in particular affiliative social behaviors, while c-Fos expression in PVN was not increased. Next, the authors quantified c-Fos expression in various “social” brain structures (they chose amygdala and bed nuclei of stria terminalis (BST) in addition to MPOA) after partial social isolation and found increased activity in various anxiety-related subnuclei of these structures such as the central amygdala, while no increased activity was found in the preoptic area (including MPOA). They did the same after social reunion and found strikingly different patterns of c-Fos labelling, specifically in the preoptic area, as well as some other areas such as medial

amygdala. Next, the authors showed that a probably receptor for amylin, calcitonin receptor (Calcr), is expressed in MPOA as well as some other social forebrain regions such as BST and medial amygdala, revealing a general pattern that was notably similar to the c-Fos labelling after social reunion. When they performed anti-Calcr stainings of the c-Fos material of social reunion mice, they found colabelled neurons especially in MPOA, and further found that these were vGluT2-expressing. In contrast, social isolation mice did not reveal Calcr+/-Fos+ colabelled neurons. Next, the authors tested the effects of excitotoxic NMDA-lesions of MPOA on social isolation and reunion behaviors and found that biting of the separation wall in partial social isolation experiments as well as social contacts and sleep huddle in reunion experiments were reduced. To test whether this was due to Calcr-signalling in MPOA, they performed RNAinterference-knockdown of Calcr in MPOA using shRNA against Calcr introduced via an AAV. In partial social isolation experiments, this knockdown lead to attenuated “protest” behaviors against the partial isolation, as well as reduced contact-seeking behaviors after social reunion, while long-term contact behaviors and sleep huddle were not inhibited, indicating a role of Calcr MPOA neurons in acute-phase social isolation/reunion behaviors. The authors then investigated the relationship between Calcr neurons and amylin neurons. Using a novel Amylin-Cre mouse which the authors constructed, they found contrary to intuition no Amylin+-synapses close to Calcr+somata, but rather Calcr+-synapses close to Amylin+-somata, and concluded that this might be a case of retrograde signalling. The authors next checked sexual dimorphism of the amylin / Calcr system and found that males had fewer of those neurons and were less afflicted by partial isolation as well as showed more aggression upon social reunion. Ultimately, the authors performed activating-DREADD experiments in central amygdala CaMK2a neurons, finding that social motivation behavior during partial isolation was increased.

In general, this is a very interesting study full of interesting findings and beautiful data. The study offers a broad basis of methods which were elegantly used by the authors to investigate their specific scientific questions.

Response: We sincerely appreciate your careful evaluation and positive assessment of our manuscript. Texts were colored in **green** for revisions attending to your comments.

However, as my summary of the whole manuscript above already shows, parts of the study appear to be juggled together, which leads to a loss of the focus of the study. For example, the last two parts of the study (and the pertaining figures 8 and 9) seem out-of-place. I think their main results could be integrated into the manuscript at previous positions, and their figures strongly condensed (with parts moved to the supplements). Right now, the manuscript at times appears too bloated. This is also indicated by some of my following specific comments and with respect to other parts of the manuscript. I would like to propose

to the authors to revise the manuscript, put more focus on the most relevant aspects and try to integrate the not-so-relevant aspects more smoothly (or, if they do not fit, remove them from this manuscript). I am sure that after a thorough revision, this will be a very nice paper suitable for publication in Nature Communications.

Response: Thank you very much for this constructive advice. We agree that the original manuscript had too many related but not indispensable data, obscuring the main story. On top of that, we had to add five additional experiments to answer the important editor's and reviewers' comments in revision. Therefore, we have cut the following parts from the original manuscript and replaced with the new data;

Figure 8 and 9: for being out of place and not fitting to the main story.

Latter half of Figure 6 (icv): for using inferior techniques as pointed by this reviewer, and replaced with the new experiments using amylin gene-targeted line.

Figure S3, S11: not indispensable.

Also we have combined several Supplementary Figs to make them condensed. As a result, we have added 5 new experiments but reduced the number of figures (from 9 to 8 in the main figs, and from 11 to 8 in the supplement). Moreover, we believe the story is now more streamlined and coherent.

Specific comments:

- At several points, the authors distinguished between social reunion (as affiliative behavior) and “defensive huddle” (as social stress coping behavior). However, to induce defensive huddle they just exposed some mice to very bright light, which means that any effects observed probably result from general stress rather than the actual social coping behavior. All parts relating to “defensive huddle” behavior should be removed from the manuscript. This would in my opinion increase the quality of the study.

Response: We understand the reviewer's comment here and below, that any effects caused by the bright-light induced defensive huddle contains those not only caused by social contacts, but also caused by bright-light exposure itself. We agreed that our descriptions were often not correct for this sense. Meanwhile, we would like to have a control social condition for social reunion, which includes direct physical contacts with peers but in a different context. That is, neuronal activation by reunion described in Fig. 4 and 5 can be simply derived from somatosensation by physical contacts with peers if we don't compare with those after defensive huddle. And indeed, the MeAD and BSTpr are commonly activated by reunion and defensive huddle. However, the subregions in the MPOA are activated only by reunion, but not by defensive huddle, suggesting that we cannot attribute the activation of MPOA subregions simply by direct physical contacts. Therefore, while removing this part from the manuscript is itself easy, we still feel that defensive huddle is an important control experiment, even though it may not be an ideal one.

In order to address this issue, we first revised the green-colored parts to the following text;

(Original) " For comparison, here we also characterized another kind of social contact, called *defensive huddle*. Defensive huddle is observed under potential threat in mice and rats, and can be regarded as a primitive gregarious behavior or a "selfish herd" {Hamilton, 1971 #2471}, resulting from cover-seeking, centripetal movement in which each animal tries to reduce its chance of being caught by a predator. "

(Revised) " For comparison, here we also characterized a different kind of situation that involved significant physical contact among cage mates, called defensive huddle {Bowen, 2013 #2441}. Defensive huddle is defined as aggregation of individual animals as a strategy of self-defense under a potential threat. This primitive gregarious behavior is designated as a "selfish herd" {Hamilton, 1971 #2471} and thus does not have an affiliative nature toward conspecifics. " (Page 9)

We describe other changes according to the reviewer's comments below.

We would really appreciate if you could review these clarifications and changes in the present manuscript here and below, and give us your precious opinion on this matter.

- I see a small weakness in the lack of conceptual interpretation of the diverse behaviors which the authors have identified and selectively scored. While I think that the particular distinguished behaviors were well selected, I would recommend some simplification in the text: For example, I would summarize the diverse behaviors under overarching terms such as "body contact behaviors" and "social motivation behaviors" (the latter e.g. referring to all kinds of bar biting and rearing etc. in partial isolation experiments, which just indicate that the isolated mouse really wants to get through the bars back to its peers). This would make the text easier to understand in my opinion.

Response: According to this comment, we categorized the observed behaviors after somatic isolation in the main text as follows: " The mouse behaviors after the partition replacement were coded based on previously-reported mouse ethogram {Mackintosh, 1963 #2612; Barnett, 1976 #2583} as follows: 1) General movements: still (no movement), movement (locomotion or any physical movements not otherwise classified); 2) exploration of the environment: sniffing (targeted at partition or through the partition), rearing, digging; 3) contact-seeking: biting at the partition, 4) non-social behaviors: self-grooming, eating, panic. "

And for the behaviors after reunion, as follows,

" the mouse behaviors after the reunion were coded for the above-described categories 1)~4), and 5) direct social behaviors: crawling under/pushing, sniffing (peer), allo-grooming, mounting, biting (peer), chasing." The categorization was also used in the Method section.

- The Methods section in particular requires English proofreading. The whole manuscript

would in fact greatly benefit from native English proofreading.

Response: The manuscript has been proofread by a native speaker, we hope the improvements increase the readability.

- Line 35: Why conserved? Could be, but this is not really discussed in the manuscript.

Response: Sorry for this unclear phrase, we have revised the sentence as follows;

(Original) These data present a novel and conserved neuromolecular basis of mammalian affiliative behaviors

(Revised) *As the same amylin-CalcR signaling is essential for maternal care, the present data also provide neuromolecular evidence for the long-postulated common origin of social affiliation and maternal care in mammals.*

- Line 33: “recover” should probably read “recovery”

- Line 47: hypothalamus-adrenal-pituitary should read hypothalamic-pituitary-adrenal

- Line 102: delete “the”

Response: Corrected accordingly.

- Line 106: first-time mention of brain structures, provide full names

Response: Sorry for the omission, we have provided the full names.

- Line 120: The “restraint stress” paradigm is not described in the Methods section. Please add this and elaborate.

Response: Sorry for the omission, we have provided the protocol for the restraint stress in the Animals section in the Method, *“To examine the effect of restraint stress on the Amylin mRNA expression, group-housed virgin female mice were subjected to single-housing for 6 days, then they were assigned for stress+ co-housing, stress- co-housing, stress + isolation, stress – isolation groups for the next 6 days. Co-housing groups were housed as four in one cage. Stress + groups were subjected daily-2hr restraint stress using ventilated 50 ml plastic tubes during 6 days, as described {Yoshihara, 2021 #2554}. Then they were subjected for perfusion fixation and the brains were sampled.*

” (page 21).

- Line 125 ff: change to “however, neither environmental (...) were sufficient to induce (...)”

Response: Corrected accordingly.

- Line 176 ff: The term “real partition” refers to an insertable wall with bars in contrast to a wall with open doors (“sham partition”), but these terms are confusing, especially in view of the previously applied wire-cage-within-group-housing-cage paradigm. First, talking about “partition walls” instead of just “partitions” would slightly improve understandability. Second, I think that the “sham partition” concept is not relevant, as mice can freely move through the doors as if there was no barrier.

Response: According to this comment, we have replaced the term "partition" into "partition wall". (page 6, line 157). The concept of "sham partition", which is penetrable but otherwise very similar to the real partition wall, and also bitable for mice, is used to control the natural tendency to bite metals observed in mice as pointed by the reviewer #1's comment (page 6 in this file).

- Line 196 ff: The term “Somatic isolation” does not really make sense. Instead, I would suggest using a term such as “partial isolation”. This would also have to be corrected in figures, such as Fig. 1.

Response: We appreciate the suggestion. We chose the term somatic isolation to capture the fact that while the body of the subject mouse is physically separated, the exchange of sensory information is still possible. We still feel that this better describes our experimental protocol than alternatives, such as partial isolation, which have multiple interpretations. For this reason we continued to use somatic isolation in the revised manuscript, however if the reviewer and editor have strong feelings about this issue we are happy to reconsider.

- Line 200-202: Please improve English grammar.

Response: We have revised the sentence as follows;

(Original) Throughout this period, there was no statistically significant differences in the gross locomotor activities measured by an implanted acceleration sensor nor in core body temperature among groups (Fig. S4a-d).

(Revised) After 45 min, these exploratory behaviors decreased, **with no** statistically significant differences in total locomotor activity or core body temperature among the groups (Fig. S2a-d) (page 7, line 192).

- Lines 196-253 (describing results in Fig. 2) and 1216-1231 (Fig. 2 incl. legend): In great detail, behaviors resulting from complete or partial (“somatic”) isolation vs. group-housing are described. I have several problems with this part: 1) it appears to lose focus of the main findings and purpose of the study, so it should be greatly condensed and partly moved to the supplementary material. 2) Main focus of Fig. 2 and the accompanying text should be the finding that both complete and partial isolation result in stress behaviors. 3) I do not understand the relevance of the “two-stage response” to social isolation (line 1216) in the scope of this study.

Response: We tried to condense the mentioned part, and at the same time, we had to add some more details to respond to all the reviewers' comments. Please take a look at the present version, and let us know if there are still superfluous parts.

- Lines 255-286: Again, too much focus on behavioral details resulting from the reunion of previously socially isolated mice. These observations, while in themselves interesting, do not seem of relevance to the purpose of the study.

Response: We condensed this part also, but because of necessary additions, the total length remains approximately the same. We hope the readability is increased, and will attend the additional indication(s) from the reviewer.

- Line 285: The authors claim that defensive huddle involves a PVN-mediated stress reaction. However, this is a non-sequitur: It is more probably that the bright light which they used in this paradigm by itself caused the stress-induced increase in PVN activity. In any case, the whole point seems irrelevant to the purpose of the study.

Response: We agree with the reviewer's point, that the light-induced defensive huddle should contain many reactions purely caused by the light, but not by the subsequent physical contacts. Therefore, we revised the mentioned part as follows to make this point clearer:

(Original) "This result suggests that, while reunion and defensive huddle both include increase in contacts, defensive huddle, but not reunion, involves PVN-mediated stress reaction."

(Revised) "Therefore, while reunion and light-induced defensive huddle commonly involve increased physical contacts with cage mates, these two conditions have distinct features, and reunion does not include a stress response." (Page 10, lines 262).

- Lines 319-327: Same problem as previous comment, the bright light probably induced

stress which resulted in the observed c-Fos patterns, and not the “defensive huddle” behavior as claimed.

Response: Again, we are sorry for the incorrect sentence. To make this point clearer, we revised the following parts;

(Original) "To identify the regions activated by defensive huddle, "

(Revised) " To identify the regions **associated with light-induced** defensive huddle, "

(Original) " The overlap of the activation pattern by defensive huddle and reunion was found only in the BSTpr and MeAD, even though both involved a social contact increase, while the preoptic subregions, namely the ACN, cMPOA, MPNm were specifically activated by affiliative social contacts. "

(Revised) "**The BSTpr and MeAD were commonly activated after reunion and blight-light induced defensive huddle, thus these regions might be activated in response to direct physical contact. On the other hand, the preoptic subregions, namely the ACN, cMPOA, MPNm were activated only by reunion, suggesting that these regions were associated specifically with affiliative contacts.**" (Page 11, lines 301).

- Lines 430-436: To test amylin function onto Calcr neurons in social behaviors, the authors injected the Calcr antagonist AC187 into the cerebral ventricles and found some effects on social contact behaviors. However, I wonder given the many caveats this method carries (such as: does the molecule actually reach the Calcr neurons by diffusion? Which Calcr neurons in which regions are inhibited? Etc.), why the authors did not perform direct infusion of AC187 into MPOA via implanted guide cannulas as in (e.g.) Hasan, Mazahir T., et al. "A fear memory engram and its plasticity in the hypothalamic oxytocin system." *Neuron* 103.1 (2019): 133-146.

Response: We agree that icv experiments have well-known drawbacks, and we first tried direct microinfusion of AC187 into the cMPOA as the reviewer suggested. However, the implanted cannulas were very often bitten and broken by peer animals. This is an inherent difficulty of studying affiliative social behaviors in which we must keep the subject mice with peers for several days, unlike experiments with single housed animals such as fear conditioning. Therefore, we next tried amylin KO mice (Fig. 8), and along with Calcr knockdown mice (Fig. 6), these experiments consistently demonstrated that the necessity of amylin-Calcr signaling for contact seeking behaviors. Thus we replaced the AC187 i.c.v. with data from these new experiments in the revised Fig. 8.

- Lines 453-465: Why did the authors focus exclusively on CeM CamK2a neurons? I find their short justification given in the text too little. Especially, since this part of the study appears entirely disconnected from the actual topics of Amylin and Calcr.

Response: According to this and other reviewers' comments, we have removed this figure from this manuscript.

- Figure 1a: Not clear to the reader why NPI was used as second antigen in this staining

Response: We have modified this part as follows; " Anterior commissural nucleus (ACN) is characterized by the cluster of magnocellular oxytocin neurons {Castel, 1988 #667}, which was detected by neurophysin I (NPI), the cleavage product of preprooxyphysin that acts as a carrier protein of oxytocin. "

- Figure 1 legend:

o In lines 1203-1205 there is a problem, these lines need to be redacted entirely.

Response: We are sorry for the incorrect sentence. We revised the following parts;

(Original) "(k, l) Distribution of amylin (magenta) and c-Fos (green) in the MPOA in relation to the co-housing or 2 hr-isolation. Arrows indicate double-labeled cells. scale bars, 50 μ m (m-n) Percentage of amylin-ir neurons expressing c-Fos, percentage of c-Fos-ir neurons expressing Amylin, the number of amylin-ir neurons and the number of c-Fos-ir neurons in the MPOA 2 hours after co-housing or 2isolation. n=4 per group."

(Revised) " (k) Top: group-housed female mice were isolated for 2 days and then cohoused for 2 h or kept isolated. Bottom: female mice housed in group were isolated for 2 h or kept in group. (l) Distribution of amylin-ir (magenta) and c-Fos-ir (green) neurons in the MPOA. Arrows: double-labeled cells. Scale bar, 50 μ m. (m, n) Number of amylin-ir and c-Fos-ir neurons, percentage of amylin-ir neurons among the c-Fos-ir neurons, percentage of c-Fos-ir neurons among the amylin-ir neurons in the MPOA ($n = 4$ mice per group). "

In general, there are many small grammar problems in this legend, please proofread.

Response: We have modified this legend and the manuscript has been proofread by a native speaker. Thank you very much again for your precious advice and comment.

Other changes: (Major changes were colored in orange-brown)

- 1 We omitted the term "sexual dimorphism", as it was pointed that this construct is not a mere difference between males and females.
- 2 The reference 47 in the original manuscript, Yoshihara et al, has now been published and the relevant text was changed accordingly.
- 3 According to the reviewers' suggestion, we have proofread the manuscript and also condensed. Minor changes caused by this procedure and do not affect the contents were left in black.

Reviewers' Comments:

Reviewer #1:

Remarks to the Author:

I commend the authors for the careful response to the review comments, the introduction of new experiments, and the reformatting of the manuscript. This is fascinating work, and it will be an important addition to the literature. I strongly support the publication of this manuscript, following a few minor changes.

Minor text changes

Line 27: change "Here we have report" to "Here we report"

Line 30: change "interaction" to "interactions"

Line 96: change "modestly" to "only moderate expression levels are observed"

Figure panel changes

Figure 4A. These immunofluorescence images identify the anatomical boundaries which were used to quantify c-Fos expression. However, the authors have chosen not to show any c-Fos immunofluorescence images in this Figure.

I would like to see in Figure 4A representative c-Fos images incorporated for the two conditions (3-together, and somatic isolation), side by side to the sections the authors have already illustrated. The rest of the Figure works well and clearly communicates the intended message.

Figure 5. For panels b, d, e, and f, place the titles of the graphs as y-axis labels.

Figure 7. For panel g, place the titles of the histograms as y-axis labels.

Reviewer #2:

Remarks to the Author:

The manuscript submitted by Fukumitsu et al has undergone substantial revision. By adding several experiments, the authors have now significantly strengthened the case for the model they propose and solidified weak links. The text has also been improved. I do not have any further suggestions for changes to the ms. In my opinion, the study, as presented in the revised ms., offers an important addition to our understanding of the neurochemical basis for social behaviour. As noted before, I also believe the models used in the study will be of interest to many in the field and adopted in subsequent studies.

Reviewer #3:

Remarks to the Author:

I found that all raised issues were very satisfactorily addressed by the authors.

I have a few further minor comments on the revised manuscript. Once these issues are addressed, this will be a beautiful paper published in Nature Communications.

- Introduction, line 78:

"The medial..." should start with: "We found that the medial..."

Also, "are" should read "area".

- Introduction, line 90:

The end of the introduction lacks some further summary of key findings of the study, as a link towards results section. I suggest:

"... social housing conditions,..." End this sentence with full stop after "conditions". Then add summary of further key findings, such as:

"When we dissected the underlying neuronal system, we found by means of genetic knockdowns, amylin agonist infusion, and chemogenetic DREADDs experiments that amylin-Calcr signalling in the cMPOA is required for social contact seeking during reunion after social isolation episodes. This suggests an important wider role of MPOA amylin-Calcr circuits in affiliative social behaviors."

The authors may adapt this according to your own judgement. If they strongly feel that it would not be necessary, I would not see it as an obligatory addition for publication. However, I find that it would put more focus on the impact of the paper.

- Results, line 138:
Correct error "or, or"

- Results, line 403:
"and then subjected", replace with "and were then subjected".

- Discussion, line 446:
"activate" should read "activated".

- Discussion, line 485:
"has" should read "have".

- Discussion, line 486:
"has" should read "have".

- Discussion, line 497:
Remove "the".

Referees' comments and our point-by-point responses

*Please note that, in the revised main text, we used Microsoft Keep Track function.

Reviewer #1 (Remarks to the Author):

I commend the authors for the careful response to the review comments, the introduction of new experiments, and the reformatting of the manuscript. This is fascinating work, and it will be an important addition to the literature. I strongly support the publication of this manuscript, following a few minor changes.

Minor text changes

Line 27: change "Here we have report" to "Here we report"

Line 30: change "interaction" to "interactions"

Response: Thank you very much for your kind support. We have corrected above changes as indicated.

Line 96: change "modestly" to "only moderate expression levels are observed"

Response: For the consistency with the preceding sentence, we have modified the mentioned term as follows; "amylin is robustly expressed in the cMPOA and the anterior commissural nucleus (ACN) and is only moderately expressed in medial part of the medial preoptic nucleus (MPNm)".

Figure panel changes

Figure 4A. These immunofluorescence images identify the anatomical boundaries which were used to quantify c-Fos expression. However, the authors have chosen not to show any c-Fos immunofluorescence images in this Figure.

I would like to see in Figure 4A representative c-Fos images incorporated for the two conditions (3-together, and somatic isolation), side by side to the sections the authors have already illustrated.

The rest of the Figure works well and clearly communicates the intended message.

Response: According to this comment, we have added a new supplementary Figure 2i. We

could not make it fit into the main figure, because in Figure 4A we would like to emphasize the comparisons between somatic isolation vs. reunion, along with those between 3-together vs. somatic isolation, spanning multiple coronal planes. Putting all of these panels will make Figure 4A too large, and exceed the journal style indication.

Figure 5. For panels b, d, e, and f, place the titles of the graphs as y-axis labels.

Figure 7. For panel g, place the titles of the histograms as y-axis labels.

Response: We have modified these panels accordingly.

Reviewer #2 (Remarks to the Author):

The manuscript submitted by Fukumitsu et al has undergone substantial revision. By adding several experiments, the authors have now significantly strengthened the case for the model they propose and solidified weak links. The text has also been improved. I do not have any further suggestions for changes to the ms. In my opinion, the study, as presented in the revised ms., offers an important addition to our understanding of the neurochemical basis for social behaviour. As noted before, I also believe the models used in the study will be of interest to many in the field and adopted in subsequent studies.

Response: We are deeply honored and rewarded by your kind words.

Reviewer #3 (Remarks to the Author):

I found that all raised issues were very satisfactorily addressed by the authors. I have a few further minor comments on the revised manuscript. Once these issues are addressed, this will be a beautiful paper published in Nature Communications.

- Introduction, line 78:

“The medial...” should start with: “We found that the medial...”

Also, “are” should read “area”.

Response: Thank you very much for your kind support, throughout this revision procedure. We have corrected above changes as indicated.

- Introduction, line 90:

The end of the introduction lacks some further summary of key findings of the study, as a link towards results section. I suggest:

“... social housing conditions,...” End this sentence with full stop after “conditions”. Then add summary of further key findings, such as:

“When we dissected the underlying neuronal system, we found by means of genetic knockdowns, amylin agonist infusion, and chemogenetic DREADDs experiments that amylin-Calcr signalling in the cMPOA is required for social contact seeking during reunion after social isolation episodes. This suggests an important wider role of MPOA amylin-Calcr circuits in affiliative social behaviors.”

The authors may adapt this according to your own judgement. If they strongly feel that it would not be necessary, I would not see it as an obligatory addition for publication. However, I find that it would put more focus on the impact of the paper.

Response: Thank you very much for the advice. We have added the suggested summary with minor modifications to fit into the context as follows;

"While studying Calcr+ and amylin+ neurons with regard to maternal care, we observed that amylin expression in MPOA subregions is heavily dependent on social housing conditions in female mice. Therefore, in this study we dissected the molecular and circuit underpinnings of these socially induced changes in female mice by means of genetic knockdowns, amylin infusion, and chemogenetic DREADDs experiments, and found that amylin-Calcr signaling in the cMPOA is required for contact seeking behavior in an affiliative context."

- Results, line 138: Correct error “or, or”
- Results, line 403: “and then subjected”, replace with “and were then subjected”.
- Discussion, line 446: “activate” should read “activated”.
- Discussion, line 485: “has” should read “have”.
- Discussion, line 486: “has” should read “have”.
- Discussion, line 497: Remove “the”.

Response: We have corrected these errors accordingly. Please accept our gratitude for your careful reading and suggestions!